



# Investigating emission sources and transport of aerosols in Siberia using airborne and spaceborne LIDAR measurements

Antonin Zabukovec[1], Gerard Ancellet[1], Iwan E. Penner[3], Mikhail Arshinov[3], Valery Kozlov[3], Jacques Pelon[1], Jean-Daniel Paris[2], Grigory Kokhanenko[3], Yuri S. Balin[3], Dmitry Chernov[3], and Boris D. Belan[3]

[1]Laboratoire Atmosphere Milieux, Observations Spatiales (LATMOS), CNRS, Sorbonne Université, Université Versailles St Quentin, Paris, France
[2]Laboratoire des Sciences du Climat et de l'Environnement, CEA-CNRS-UVSQ, Gif sur Yvette, France
[3]V.E. Zuev Institute of Atmospheric Optics, Russian Academy of Sciences, Siberian Branch (IAO SB RAS), Tomsk, Russia

**Correspondence:** Antonin Zabukovec (antonin.zabukovec@latmos.ipsl.fr)

**Abstract.** Airborne backscatter lidar measurements at 532 nm were carried out over Siberia in July 2013 and June 2017. The Russian Tu-134 flew over major Siberian cities (Novosibirsk, Tomsk, Krasnoyarsk, Yakutsk), the gas flaring fields of the Ob valley and Siberian Taiga in order to sample several kinds of Siberian aerosol sources. Aerosol types are derived using the Lagrangian FLEXible PARTicle dispersion model (FLEXPART) simulations, Moderate Resolution Imaging Spectrometer (MODIS) Aerosol Optical Depth (AOD), Infrared Atmospheric Sounding Interferometer (IASI) CO total column and AOD at 10 μm. Forest fire detection is based on NASA Fire Information for Resource Management System (FIRMS) from MODIS and the Visible Infrared Imaging Radiometer Suite (VIIRS) observations and airborne in-situ measurements when available. Six aerosol type could be identified in this work: (i) Dusty aerosol mixture (ii) Ob valley industrial emission (iii) fresh boreal forest fire plumes (iv) aged forest fire plumes (v) pollution over the Tomsk/Novosibirsk region (vi) long range transport of Chinese pollution over Yakutsk. The backscatter to extinction ratio and then the corresponding lidar ratio (LR) were derived for each of these 6 identified aerosol type, using an iterative method based on the Fernald forward inversion constrained by the 10 km MODIS collection 6 AOD distribution closed to the airborne lidar observation. The LR analysis showed that the lowest LR range was obtained for the "Dusty Mix" case (26-40 sr) and the highest for the urban and industrial pollution from the Tomsk/Novosibirsk area (71-90 sr). The comparison is good with previous estimate of LR according to the aerosol classification. The range of lidar ratio obtained for gas flaring emission (43-60sr) was lower than the high values encountered in the Tomsk/Novosibirk urban area and has never been characterized using lidar observations. Airborne lidar backscatter ratio vertical structure, aerosol types and integrated LR derived from the airborne data analysis were compared to nearby CALIOP overpasses. These comparisons showed three main differences with the CALIOP LR and aerosol type classification over Siberia: (i) CALIOP aerosol layer can be classified as Elevated smoke instead of Polluted continental and vice versa, but with little influence on the LR value (ii) aging and transport of aerosol layers effect on the CALIOP LR value is not always properly accounted for even when the CALIOP classification is correct (iii) the lack of discrimination between fresh and old fire plume leads to an overestimation of the optical depth for the fresh fires in the CALIOP AOD over the fire source region.





# 1 Introduction

Atmospheric aerosols play a very important role in many meteorological, radiative and chemical processes taking place in the
atmosphere such as cloud formation, scattering and absorption of incident solar (short-wave) and thermal (long-wave) radiation
from the Earth, as well as affect the air quality (Chỳlek and Coakley, 1974). Due to the variety of the optical, microphysical
and chemical properties of atmospheric aerosols, closely depending on the formation processes of particulate matter and their
subsequent aging processes occurring in the atmosphere, and the poor knowledge of their spatio-temporal distribution as well,
they have been identified by the Intergovernmental Panel on Climate Change (IPCC) as one of the main uncertainty sources
when assessing radiative forcing and the climate change (Stocker et al., 2013).

Siberia has been widely recognized as a large source region of biomass burning aerosols, the impact of which on the aerosol
load is well identified (Lavoué et al., 2000; Paris et al., 2009b). Regarding anthropogenic aerosol sources, Asian pollution and
gas flaring from oil wells in Siberia have been identified as key aerosol sources (Stohl et al., 2013), however the impact of
these pollutants is underestimated largely due to the lack in reliable data on Russian emissions (Bond et al., 2013; Huang et al.,
2015).

The YAK-AEROSIB project demonstrated that airborne measurements of atmospheric concentrations of $CO_2$, $CH_4$, CO,
$O_3$ and aerosol content in Siberia are very valuable to identify the anthropogenic and natural sources of aerosol and trace
gases (Paris et al., 2008, 2009a). Lidar measurements of the aerosol vertical distribution in Russia have been also reported by
Dieudonné et al. (2015) using a 355 nm mobile backscatter lidar installed in a van making a road transect between Smolensk
(32°E, 54°N) and Lake Bailkal (107°E, 51°N). A dust outbreak near 70°E and the ubiquity of biomass burning plumes have
been identified as the major results about the aerosol distribution in Siberia during this field experiment. Regular lidar obser-
vations have been made in Siberia in the city of Tomsk using either a ground based multiwavelength Raman lidar to make
nighttime profiles from March to October (Samoilova et al., 2010, 2012), or 18 months of daily measurements with a 808
nm micropulse lidar coupled to the Tomsk sunphotometer (Ancellet et al., 2019). Samoilova et al. (2012) has shown that the
optical characteristics of the aerosol lidar measurements (angstrom coefficient and lidar ratio) can be well explained using an
urban aerosol model and that the seasonal variability (difference between the warm and cold months) is weak in the planetary
boundary layer (PBL) but significant in the free troposphere (FT). Aerosol type seasonal variability and sources in Siberia
derived by Ancellet et al. (2019) showed that 56% of the detected aerosol layers are linked to natural emissions (vegetation,
forest fires and dust) and 44% to anthropogenic emissions (one-third from flaring and two-thirds from urban emissions). Since
these results are mainly related to observations near the Siberian cities, airborne lidar measurements at the regional scale are
also needed to get a better insight in the aerosol sources and transport in Russia. Airborne lidar campaigns conducted elsewhere
in the world have been indeed very valuable to characterize the regional distribution of aerosol sources, e.g. in North America
(Burton et al., 2012, 2013), in Europe and North Africa (Groß et al., 2013), or the Indian Ocean (Pelon et al., 2002). Errors in
the aerosol layer optical properties retrieval from lidar observations can be largely ascribed to incorrect aerosol classification
or incorrect lidar ratio (LR)(Rogers et al., 2014). The airborne lidar data analyzed in this paper for the campaigns conducted in
Russia will focus on the retrieval of these two parameters.





While airborne or ground based lidar measurements only provide the characterization of few case studies, only space-borne instruments have the capability to provide daily global coverage of the Earth with a good spatial resolution. The Cloud-Aerosol Lidar with Orthogonal Polarization (CALIOP) instrument is part of the Cloud-Aerosol Lidar and Infrared Pathfinder Satellite Observations (CALIPSO) mission (Winker et al., 2009). This platform was launched in April 2006 as part of the A-train constellation. CALIOP provides attenuated backscatter signal at 532 nm and 1064 nm and depolarization at 532 nm. In the CALIOP aerosol data processing scheme, the aerosol classification is essential to accurately determine the aerosol extinction and optical thickness (Omar et al., 2009; Kim et al., 2018), since the extinction to backscatter ratio (lidar ratio) depends on the aerosol type, age and mixture. Regional aerosol studies with CALIOP have been conducted for high latitudes (Pierro et al., 2011; Devasthale et al., 2011), European Arctic (Ancellet et al., 2014), or the Arctic ice sheet (Di Biagio et al., 2018), but similar studies is also needed for Siberia. Regional airborne lidar campaigns in Siberia are then essential for future analysis of the CALIOP observations in this region.

In this paper we present the analysis of the data obtained during two different aircraft campaigns conducted in Central and Eastern Siberia in 2013 and 2017 with a Tu-134 aircraft equipped with in-situ trace gas and aerosol sensors as well as a downward looking 532 nm elastic-backscatter lidar. The analysis of the lidar ratio is however limited by the lack of high spectral resolution lidar (HSRL) or Raman detection capabilities for the Russian lidar on-board the Tu-134 aircraft, so only aerosol layers with simultaneous aerosol optical depth observations by Moderate Resolution Imaging Spectroradiometer (MODIS) at a 10-km horizontal resolution will be considered in this work (Royer et al., 2010; He et al., 2006). The aircraft campaigns,the instruments and the methodology to identify the aerosol type and to process the lidar data are described in section 2 and 3. In section 4, six case studies are analyzed in term of aerosol type encountered and optical properties derived, while the comparison with previous LR analysis is made for the types of aerosols identified in this study. Finally a selection of CALIOP profiles that can be compared to the airborne lidar measurements, is analyzed in section 5 to discuss the representativeness of the aerosol type classification and lidar ratio values from CALIOP Version 4 aerosol data products over Russia.

## 2 Description of the aircraft campaign and the data set

### 2.1 Campaign description

Research aircraft operated by IAO SB RAS has been flying along the transcontinental routes over central and eastern Siberia since 2006. Two airborne campaigns took place in July 2013 and June 2017 with a backscatter lidar and *in-situ* aerosol instruments installed on board the aircraft. The four flight tracks for each campaign are shown in Fig 1.

The flights were performed over (i) the major large Siberian cities (Novosibirsk, Tomsk, Krasnoyarsk, Yakutsk) (ii) the gas flaring fields of the Ob valley and the industrial city of Norilsk (iii) as well as over the Siberian taiga in order to track the long-range transport of emissions from wildfires and mid-latitude Eastern Asia. So the major aerosol sources could be included in this analysis. During these two airborne campaigns, downward looking backscatter lidar measurements were carried out but also in-situ measurements of trace gas concentrations and aerosol particle properties (size distribution, scattering properties).

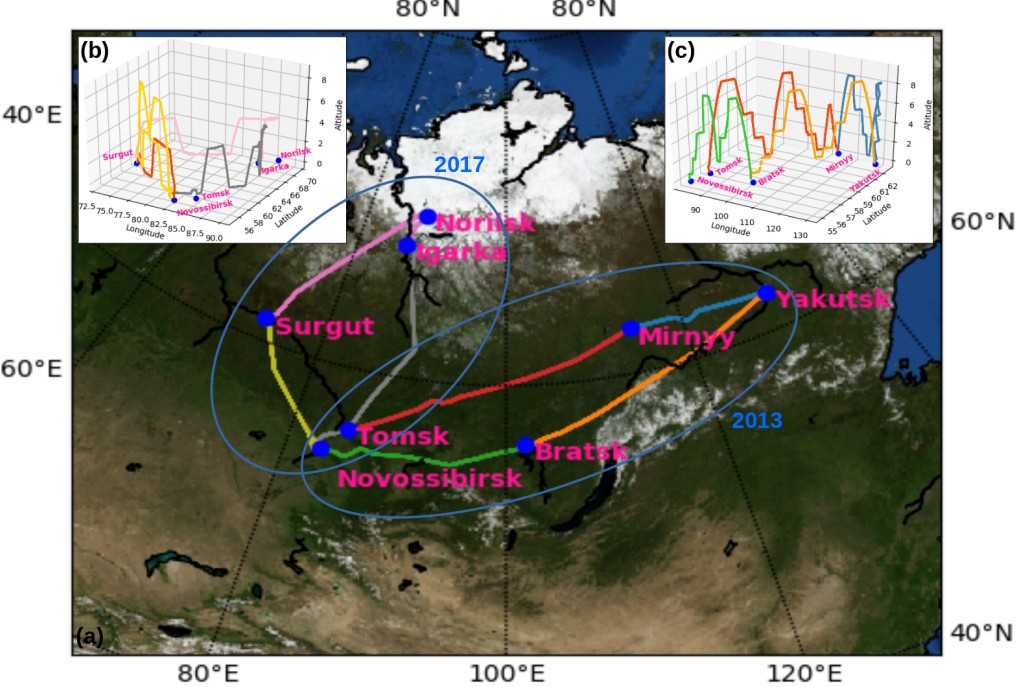

**Figure 1.** Map of the 8 aircraft flight tracks carried out in 2013 (eastern loop) and in 2017 (northern loop). The aircraft altitude ranges are also shown for the 2017 (b) and 2013 (c) flights. Map background : NASA's Earth Observatory.

The aircraft performed various legs at low altitude ($\approx 0.6$ km) and between 4 and 8 km in order to sample different atmospheric
layers. Only lidar data collected during flights above 4 km are used for the aerosol layer characterization.

## 2.2 Airborne lidar system

The lidar system installed on board the Tu-134 aircraft is based on the LOSA aerosol lidar developed at the IAO SB RAS (Balin
et al., 2011; Penner et al., 2015). The transmitter module is based on a solid state Nd-YAG laser emitting 8 ns laser pulses at
1064 nm and 532 nm. The maximum output energy at 532 nm is 100 mJ with a repetition rate of 10 Hz and a beam divergence
of 2.5 mrd. The optical receiver is a 150 mm diameter reception lens coupled with a 1nm filter and two reception channels (co-
and cross-polarization). The full geometrical overlap is obtained between 80m and 150m. In practice the first 200 m are not
used to reduce the errors when estimating the overlap function correction in clear air region below the aircraft. The detection
unit is composed of a photomultiplier coupled to an analog-to-digital converter (ADC) electronic system with a sampling rate
range of 25 - 100 MHZ (i.e. a 1.5-6m vertical resolution) and a resolution of 12 bits. The cross-polarization calibration is not
sensitive enough to characterize the aerosol type and is mainly used to discriminate cloud and aerosol layers. The near infrared
channel was not available during the aircraft campaigns. Attenuated backscatter signal in decimal logarithm $> -2.3$ with a
signal above detection threshold in the depolarization channel is considered as cloud. The initial lidar data temporal averaging





is 8 s and 1 minute, respectively for 2013 and 2017 campaign. After cloud clearing data are averaged over 1-5 minutes to get a measurement range higher than 6 km.

## 2.3 *In-situ* measurements

In-situ measurements include trace gas and aerosol measurements. CO measurement is performed using a fully automated CO analyzer based on a commercial infrared absorption correlation gas analyzer (Model 48C, TEI Thermo Environment Instruments, USA). The instrument is described in (Nedelec et al., 2003). The accuracy is 5 ppb (5 % CO ) for a 30 s integration time (i.e. the response time of the instrument) and the detection limit is 10 ppb.

$CO_2$, $CH_4$ and $H_2O$ measurements were performed by means of the Picarro G2301-m gas concentration analyzer with a 5 s precision of 70 ppb for $CO_2$ and 0.5 ppb for $CH_4$. Water correction software automatically reports dry gas mole fractions. Equivalent black carbon (EBC) mass concentration are measured using an aethalometer based on light attenuation by particles after collection on a filter (Panchenko et al., 2000). The wavelength range between 0.4 and 1.1 μm with a maximum near 0.9 μm. This instrument is sensitive to submicron particles. BC mass concentration ($\rho_{BC}$) in μg/m$^3$ is converted from light absorption measurement ($\ln \frac{I}{I_0}$) with the following relationship : $\rho_{BC} = 697 \cdot C_f \cdot \ln \frac{I}{I_0}$, where $C_f$ is the correction factor between 0.5 and 1 taking into account the blackening of the filter. The EBC sensitivity is $\simeq 0.01$ μg/m$^3$.

The particle size distribution is fully characterized using two different instruments. Ultra fine particles concentration in the diameter range from 3 to 200 nm are measured using a diffusional particle sizer (DPS) consisted of an 8-channel automated diffusion battery (synthetic screen ADB; designed by ICKC SB RAS, Novosibirsk; Ankilow et al. (1991); Ankilov et al. (2002b, a)) coupled with a condensation particle counter (TSI CPC 3781). One scanning period of the DPS takes 80 s to derive size distribution in 20 size bins. Transmission efficiency for the airborne instrument is corrected for and is ≈0.997 in the 70-200 nm range and between 0.82 at 400 hPa and 0.89 at 1000 hPa for the 3-70 nm size range. All concentrations are reported at standard pressure and temperature (STP) conditions. Particle concentrations in 31 size bins in the range from 0.25 to 32 μm are measured using a GRIMM 1.109 optical particle counter (GRIMM Aerosol Technik GmbH & Co. KG, Germany).

Meteorological parameters such as temperature, humidity and wind vector are measured routinely on-board using HYCAL sensor model IH-3602-C of Honeywell Inc. Temperature and relative humidity accuracies are 0.5ºC and 7 %, respectively.

## 3 Methodolgy of the aircraft data analysis

### 3.1 Aircraft data processing

The lidar calibration is performed several times during each flight using a normalization of the attenuated backscatter signal (PR2) to molecular backscatter in the range 200m-700m below the aircraft with a 1-5 min temporal resolution. Flight sections without aerosol/cloud occurence are determined using the in-situ aircraft measurements (total aerosol concentrations from the Grimm instrument $< 15$ particles.cm$^{-3}$) . The vertical profile of molecular backscatter was estimated from the 0.75º ERA-Interim ECWF meteorological analysis (Dee et al., 2011). To account for the the overlap function between the field of view of





the laser and the telescope, the mean altitude dependency of the ratio between PR2 and the molecular backscatter is determined

for each flight in the 0-700 m altitude range below the aircraft using only profiles with no cloud and aerosol layers . The calibration accuracy is then of the order of 5-10% due to the mean signal statistical uncertainty ($< 3\%$) and the assumption on the reference scattering ratio being unity in the calibration range ($\approx 5\%$).

Aerosol optical depth retrieval is based on the Fernald forward inversion of the calibrated PR2 (Fernald, 1984), assuming a range independent value of aerosol lidar ratio (LR). The LR value is constrained using the distribution of 10 km MODIS

collection 6 (Levy et al., 2013) Aerosol Optical Depth (AOD) in an area of $\pm$ 70 km and a time lag of $\pm$ 5 h around the aircraft observation. The possible LR values are obtained by an iterative analysis using the $25^{th}$ and $75^{th}$ percentile of the MODIS AOD distribution to constrain the lidar AOD. To estimate the uncertainty of the retrieved backscatter ratio, 500 inversions were performed using random LR values within the interval deduced from the constraint with MODIS and random backscatter coefficient at the reference altitude within the interval corresponding to the statistical error on the attenuated backscatter at the

reference altitude ($\approx 5\%$). The complete methodology for the lidar data analysis is summarized in the upper panel of Fig. 2.

## 3.2   Identification of the aerosol sources

Aerosol types of the layers observed by the airborne lidar were characterized using first the Lagrangian FLEXible PARTicle dispersion model (FLEXPART). FLEXPART is a Lagrangian model designed for computing the long-range transport, diffusion, dry and wet deposition, of air pollutants or aerosol particles backward or forward from point sources using a large

number of particles (Stohl and Seibert, 1998; Stohl et al., 2002). For our study particle dispersion calculations are performed by including aerosol tracer removal processes by dry and wet deposition in the cloud and under the cloud. For each aerosol layer identified in a lidar profile, 5-10 days backward simulations of the spatial distribution of 10000 particles released in a 1 km thick altitude zone is made. The occurrence of clouds is calculated by FLEXPART using the relative humidity fields. The meteorological fields used for the simulations (including precipitation rates) are operational ECMWF field at T255 horizontal

resolution ($\approx 80$ km) and with 153 model vertical levels. FLEXPART simulations provide maps of Potential Emission Sensitivity (PES) for each aerosol layer observed by the airborne lidar. These maps represent the areas that have most influenced the observations including the correction due to the losses of the tracer (Seibert and Frank, 2004). AOD maps from the Level-3 MODIS Atmosphere Daily Global Gridded Product ($1^{\circ}$x$1^{\circ}$ resolution) are also used to identify elevated aerosol sources in the FLEXPART source region.

Various satellite observations are then used in the source region for a first guess of the aerosol type attribution. The sources of biomass-burning aerosol are identified using the daily fire radiative power (FRP) maps based on NASA Fire Information for Resource Management System (FIRMS) from MODIS observations (Giglio et al., 2003) and the Visible Infrared Imaging Radiometer Suite (VIIRS) (Schroeder et al., 2014). Significant biomass burning aerosol production is taken into account only if the daily FRP is higher than 0.3 GW and the fire lifetime higher than three days and if the CO tropospheric column measured

by IASI is higher than the monthly background CO column (Clerbaux et al., 1998; Hadji-Lazaro et al., 1999; Hurtmans et al., 2012) (Data set Clerbaux (2018)).





Dust outbreaks from the Eastern Asia desert are only taken into account if the IASI 10 $\mu m$ AOD is greater than 0.08 in the area with elevated PES (Peyridieu et al., 2013; Capelle et al., 2014, 2018) and when CALIOP aerosol depolarization ratio is greater than 15% in the same area (Tesche et al., 2011; Groß et al., 2011, 2013). Urban pollution aerosol sources are considered only when large cities (> 500000 inhabitants) are included in the high PES area. The location of flaring sources is based on the anthropogenic emissions ECLIPSEv4 data-set (Evaluating the Climate and Air Quality Impacts of Short-Lived pollutants) described in Klimont et al. (2017). This inventory includes in particular the gridded methane emissions from gas flaring in the Russian Arctic at a 0.5° x 0.5° horizontal resolution. A threshold of 50 $\mathrm{moles.km^{-2}}$/hour has been applied to the methane emissions to select areas that could potentially be defined as flaring sources. Elevated CO tropospheric column (> $2.10^{18}$ $\mathrm{molecule.cm^{-2}}$) is also mandatory to check the contribution of industrial and urban combustion aerosol sources (Wang et al., 2018).

In addition to this first guess for the aerosol type identification, in-situ aircraft measurements of CO concentration, black carbon (BC) mass concentration and aerosol size distribution are also analyzed for aircraft ascent or descent across the aerosol layer observed by the airborne lidar. Excess of CO ($\Delta$CO), i.e. the difference with the background CO concentration taken as the minimum of CO measured during the two campaigns in the lower troposphere (0-5 km), must reach 30 ppb for biomass burning aerosol and gas flaring emission (Paris et al. (2009b)). Black carbon mass concentrations is also used to identify combustion aerosol: BC > 0.5 $\mathrm{\mu g.m^{-3}}$ and BC maximum correlated with elevated $\Delta$CO. The ratio of the aerosol concentration in the cloud condensation nuclei (CCN) mode between 15 nm and 80 nm over the aerosol concentration in the Aitken nuclei (AN) mode between 80 nm and 200 nm is used to identify the aerosol aging (Willeke and Whitby, 1975; Bäumer et al., 2008; Furutani et al., 2008). The complete methodology for the aerosol type identification is summarized in the upper panel of Fig. 2.

## 4 Aircraft campaign data analysis

Six flights have been selected during the 2013 and 2017 campaigns, because they correspond to different aerosol sources. The lidar vertical cross-section of the calibrated attenuated backscatter (PR2) have been used to identify the horizontal and vertical extent of the aerosol layers (532 nm scattering ratio larger than 1.5). For each aerosol plumes, the AOD and the integrated LR are calculated, while the aerosol type is retrieved using the methodology described in section 3.

### 4.1 Dusty aerosol mixture

On June 16, 2017 the aircraft flew between Novosibirsk and Surgut above the Ob Valley. The PR2 latitudinal cross-section when the aircraft is flying at 4.2 km, shows an aerosol layer in the 0 and 2.5 km altitude range with a 150 km horizontal extent (Fig. 3). Clouds (high backscatter and high depolarization ratio) are encountered at 57.5 N when descending to a lower flight level near 500 m. The AOD and LR are calculated for a 6 min average profile shown by the red rectangle in Fig. 3. The lidar ratio range is between 26 sr and 40 sr when using nearby 25th and 75th percentiles of nearby MODIS 550 nm AOD, respectively 0.07 and 0.112. The corresponding airborne lidar AOD at 532 nm is then 0.09 ± 0.02 (Table 1).





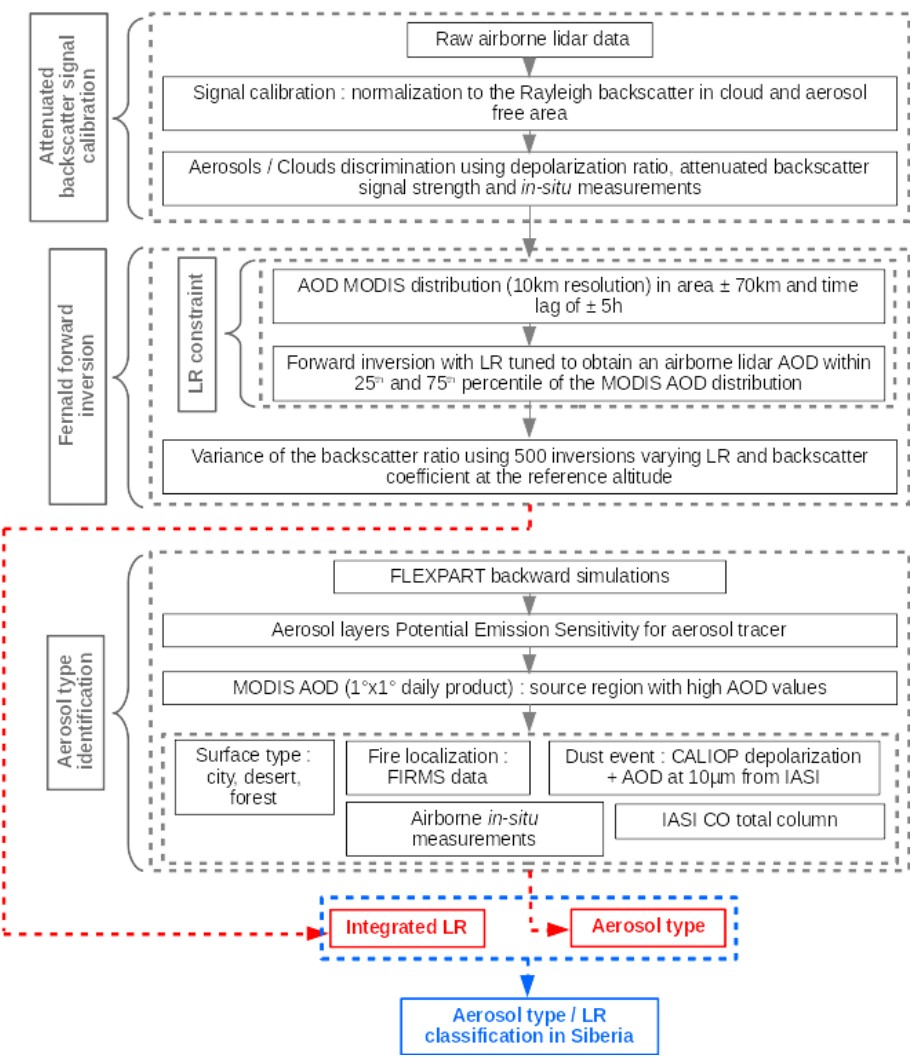

**Figure 2.** Flow chart of airborne lidar data processing and aerosol types identification.

Two areas with potential emission sensitivity (PES) higher than 2000 s for the aerosol layer in the altitude range 1 to 3 km correspond to two regions with elevated 4-day averaging of the MODIS AOD ($>$ 0.2): the Ob Valley industrial area (55°N, 80°E) and a region from 65°E-75°E above Kazakhstan at 50°N (Fig. 4). Fires are detected by MODIS and VIIRS during 3 days (FRP $<$ 0.2 GW) at 50°N, 65°E over Kazakhstan, while the tropospheric CO column measured by IASI is also high in the same area (2.0-2.5 x $10^{18}$ molecule.cm$^{-2}$). A second maximum of tropospheric CO column (1.5-1.8 x $10^{18}$ molecule.cm$^{-2}$) at 58°N,80°E (Fig. 5) is located above the industrial sources of the Ob Valley and Novosibirsk region where there is the secondary MODIS AOD maximum at 55°N.





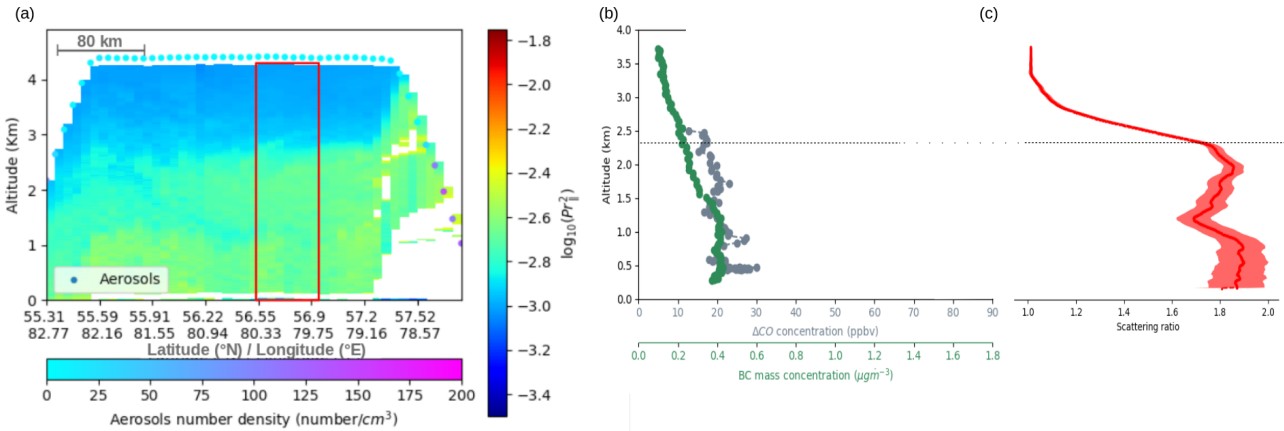

**Figure 3.** (a) : Vertical cross-section of airborne lidar $\log_{10}(PR^2)$ on June, 16 2017. Calibration constant is $13458 \pm 2\%$. Grimm aerosol concentrations in $\text{particle.cm}^{-3}$ are shown at the aircraft altitude. (b) : $\Delta CO$ and BC vertical profiles. (c) Aircraft averaged backscatter ratio vertical profile.

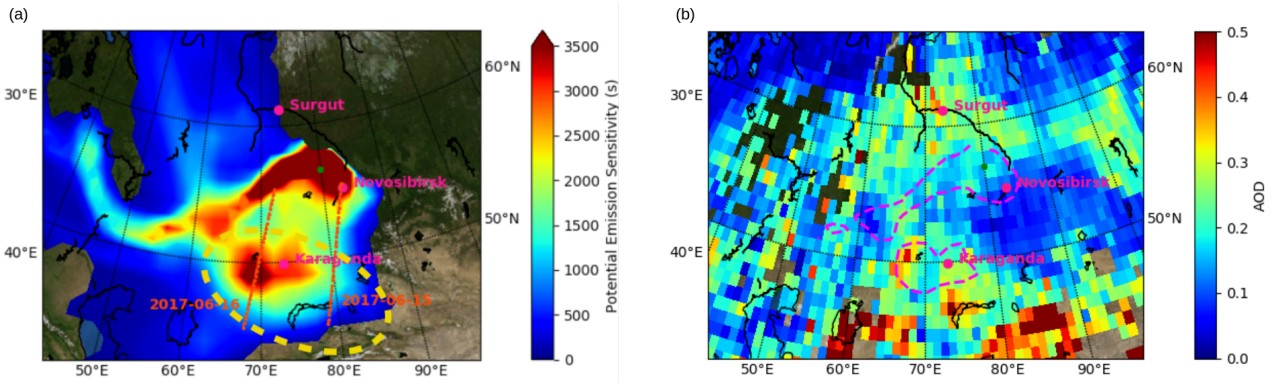

**Figure 4.** (a) : Map of the vertically integrated PES distribution from a FLEXPART backward simulation for the aerosol layers between 0-2.5 km at $56.6^{\circ}$N (green point). The orange dotted lines is the selected CALIOP overpasses in the aerosol emission area. The yellow dotted circle is the dust source area. (b) : MODIS AOD $1^{\circ}$x$1^{\circ}$averaged over the 4 days before the flight (June 13 to 16 2017) with high PES area (PES $\geq$ 2000 s) shown by the pink dotted line. The green point is the airborne lidar position. Map background : NASA's Earth Observatory.

The role of dust emission is also significant for this case study when looking at the 4-day average of the 10 $\mu m$ AOD measured by IASI which is in the range 0.15-0.35 above Kazakhstan (Fig. 5). A CALIOP overpass on June 16 also shows high aerosol depolarizing ratio (15%-20%) up to 3 km altitude at $50^{\circ}$N,$70^{\circ}$E above Kazakhstan (Fig. 5). An aerosol depolarization ratio less than 20% is consistent with polluted dust aerosol (Tesche et al. (2011); Groß et al. (2011); Burton et al. (2013); Groß et al. (2013)). In the flight area, IASI also detected a 10 $\mu m$ AOD between 0.08-0.1, which is smaller than above Kazakhstan, but

reaches the dust detection threshold of 0.08. The aerosol layer observed by the airborne lidar at $57^{\circ}$N,$80^{\circ}$E can be considered as a mixture of dust, industrial pollution the Ob Valley and aged smoke.





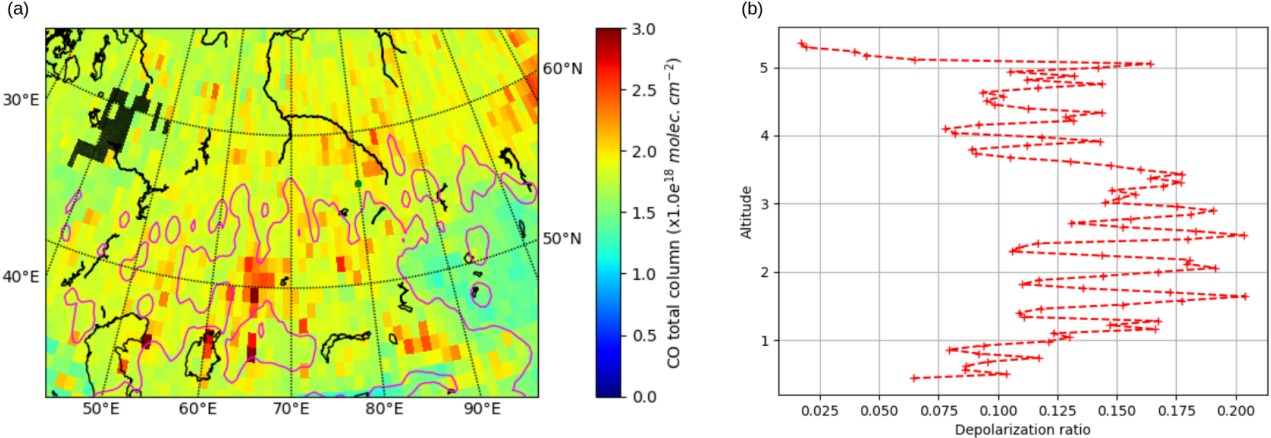

**Figure 5.** (a) : Tropospheric CO total column averaged over 4 days before the flight (June 13 to 16 2017). The pink area is the area with dust AOD > 0.08 (AOD at $10\mu m$ from IASI). (b) : Averaged CALIOP 532 nm aerosol depolarization ratio vertical profile in the aerosol source region at 49.7°N (Fig. 4a). Map background : NASA's Earth Observatory.

When looking at the in-situ measurement made by the aircraft, $\Delta$CO range is between 20 and 30 ppbv up to 2.5 km and the BC mass concentration increases from 0.2 $\mu g.m^{-3}$ at 2 km to 0.4 $\mu g.m^{-3}$ at 500 m. These moderate values of $\Delta$CO and BC compared to other flights (Table 1) are also consistent with a dusty aerosol mixture.

## 4.2 Ob Valley gas flaring emissions

On June 18, 2017, the aircraft flew again between Novosibirsk and Surgut and aerosol layers have been observed at higher latitudes above the gas and oil extraction field of the Ob Valley at 59°N. he PR2 latitudinal cross-section when the aircraft is flying at 4.5 km, shows an aerosol layer in the 0 and 3.5 km altitude range. The AOD and LR are calculated for a 6 min average profile shown by the red rectangle in Fig. 6. The lidar ratio range is between 43 sr and 60 sr when using 0.2 and 0.26 for respectively the $25^{th}$ and $75^{th}$ percentiles of the nearby MODIS 550 nm AOD. The corresponding 532 nm AOD for the airborne lidar is then 0.23 $\pm$ 0.06 (Table 1), two times larger than the previous dusty mix case observed at 57°N two days earlier.

The corresponding PES map (Fig. 7) shows again a large fraction of the air masses above the Ob Valley (55°N-60°N, 75°E) and air masses transported from Kazakhstan, but with little influence from the fire region at 67°E. Two areas corresponding to elevated 4-day average of the MODIS AOD (> 0.2) and high PES values, are identified: the oil and gas field at 5°N, 75°E and the dust emission region above Kazakhstan at 47°N,73°E (Fig. 8). The map of the IASI tropospheric CO column averaged from June 15 to June 18 is quite similar to the map averaged from June 13 to June 16 discussed in the previous section (Fig. 10a). The CO maximum of 2.0 x $10^{18}$ molecule.cm$^{-2}$ is still above the Ob Valley at 58°N,75°E. Contrary to the previous case, the 4-day average of the 10 $\mu m$ AOD (0.06) in the aircraft flight area is below the dust detection threshold of 0.8, even though sensitivity to dust emission from Kazakhstan is still highlighted by the PES analysis. A layer with dust (aerosol depolarization





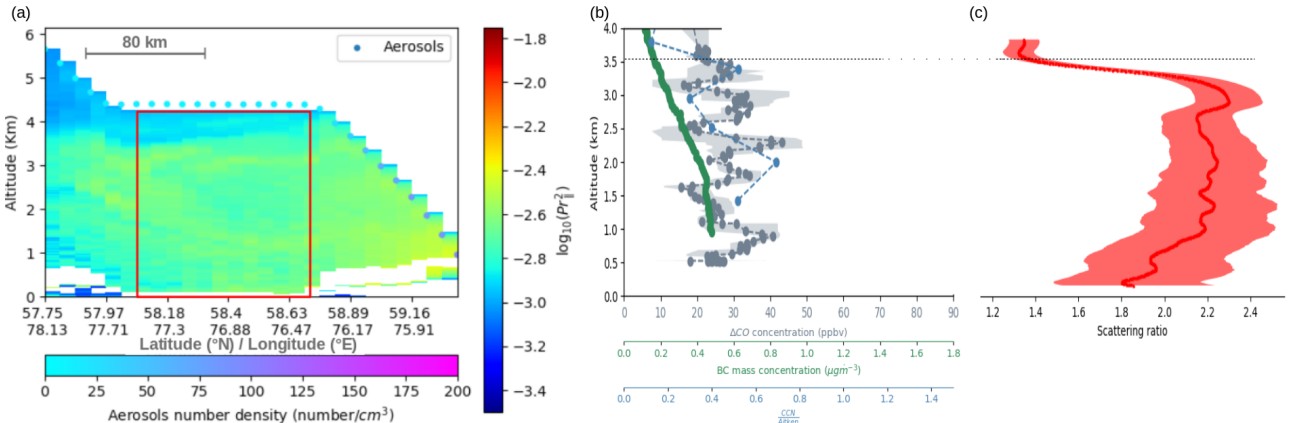

**Figure 6.** (a) : Vertical cross-section of airborne lidar $\log_{10}(PR^2)$ on June, 18 2017. Calibration constant is $63413 \pm 2\%$. Grimm aerosol concentrations in particle.cm$^{-3}$ are shown at the aircraft altitude. (b) : $\Delta$CO, CCN/Aitken and BC vertical profiles. (c) Aircraft averaged backscatter ratio vertical profile.

ratio up to 30% ) is detected by a CALIOP overpass on June 18 at the aircraft location 13 hours after the flight. However the dust layer is above an altitude of 3.5 km (Fig. 8a), so above the layer seen by the airborne lidar and the dust layer thickness is less than 1 km. Ancellet et al. (2019) already pointed out that dust layers above Tomsk (56°N,85°E) are generally detected above the boundary layer. We can conclude that the aerosol layer sampled by the airborne lidar at 59°N on June 18 below 3.5

km is then mainly related to the Ob Valley aerosol emissions and is not significantly influenced by the Kazakhstan dust and biomass burning emissions. Aircraft *in-situ* measurements during the descent at 59°N (Fig. 6) also show $\Delta$CO concentrations up 40 ppbv and BC mass concentration of 0.5 $\mu$g.m$^{-3}$ in the 1-2 km altitude range which indicate that local pollution sources have been transported up to 2 km on June 18. The moderate value of CCN to AN ratio (0.3-0.65) is also consistent with the accumulation of local emissions compared to the transport of distant emissions from the Kazakhstan deserts.

### 4.3   Fresh forest fire plume

The aircraft flew in July 2013 near a region with numerous forest fires above Northern Siberia between 60°N-65°N, 90°E-100°E. The airborne lidar detected a 500-m thick aerosol layer in the altitude range 2-4 km with a backscatter ratio of the order of 3.5 (Fig. 9). The aerosol plume sampled by the aircraft has a West-East cross section of 50 km and the AOD and LR are calculated for three 40 s average profile shown by the red rectangles at 93.2°E, 93.7°E and 94.1°E in Fig. 9. The $25^{th}$ and $75^{th}$

percentiles of the 550 nm AOD measured by MODIS in the biomass burning plume are respectively 0.115 and 0.125 in the area where the fire plume has been observed. To account for the airborne lidar not sampling 20% and 35% of the 4 km thick aerosol layer observed by MODIS at 93.7°E and 94.1°E, the 550 AOD ranges have been lowered accordingly, i.e. 0.092-0.1 and 0.075-0.081 respectively when calculating the lidar ratio at these two longitudes. For the three selected profiles, we obtain the same LR range of 32-39 sr, while the AOD measured by the lidar at 532 nm is $0.115 \pm 0.01$ for this biomass burning plume.





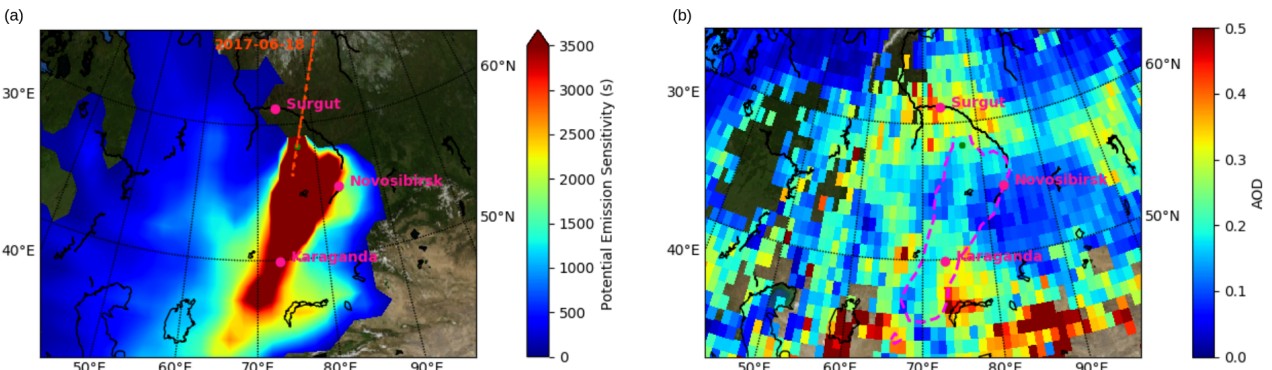

**Figure 7.** (a) : Map of the vertically integrated PES distribution from a FLEXPART backward simulation for the aerosol layers between 0-3.5 km at 58.3°N (green point). The orange dotted lines is the selected CALIOP overpasses in the aerosol emission area. (b) : MODIS AOD 1°x1°averaged over the 4 days before the flight (June 15 to 18 2017) with high PES area (PES ≥ 2000 s) shown by the pink dotted line. The green point is the airborne lidar position. Map background : NASA's Earth Observatory.

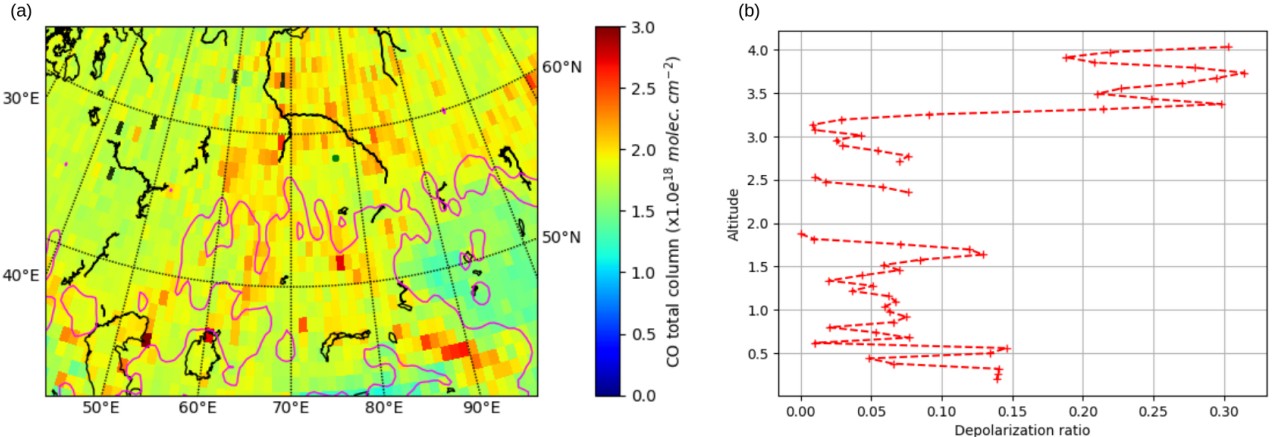

**Figure 8.** (a) : Tropospheric CO total column averaged over 4 days before the flight (June 13 to 16 2017). The pink area is the area with dust AOD > 0.08 (AOD at 10$\mu m$ from IASI). (b) : Averaged CALIOP 532 nm aerosol depolarization ratio vertical profile at 55°N, 75°E on June 18, 2017 21UT. Map background : NASA's Earth Observatory.

The PES map calculated by FLEXPART (Fig. 10) shows no aerosol transport from large cities or pollution sources. Active forest fires with a lifetime around 3 days and FRP up to 0.2 GW took place at 66°N, 108°E from June 17 to June 19 and an active fire was observed at the aircraft location (58°N, 95°E) on June 19 with a FRP of 0.18 GW and a lifetime of 1-2 days. A fresh forest fire plume is then responsible for the aerosol layer seen by the lidar with a backscatter ratio higher than 3.5. The aircraft in-situ measurements indeed show BC mass concentration between up to 1.6 $\mu g.m^{-3}$ and $\Delta$CO up 40 ppbv at 3 km,

and the low CCN to AN concentration ratio (0.1 - 0.15) is consistent with the sampling of a fresh fire plume.





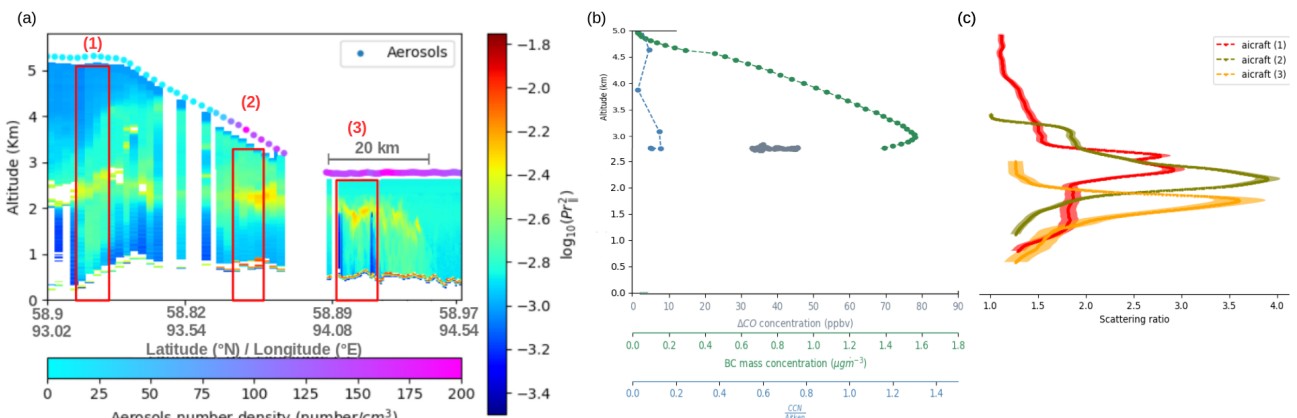

**Figure 9.** (a) : Vertical cross-section of aircraft $\log_{10}(PR^2)$ on July, 19 2013. Calibration constant is $491724 \pm 10\%$. Grimm aerosol concentrations in particle.cm$^{-3}$ are shown at the aircraft altitude. (b) : $\Delta$CO, CCN/Aitken and BC vertical profiles. (c) Aircraft averaged backscatter ratio vertical profile.

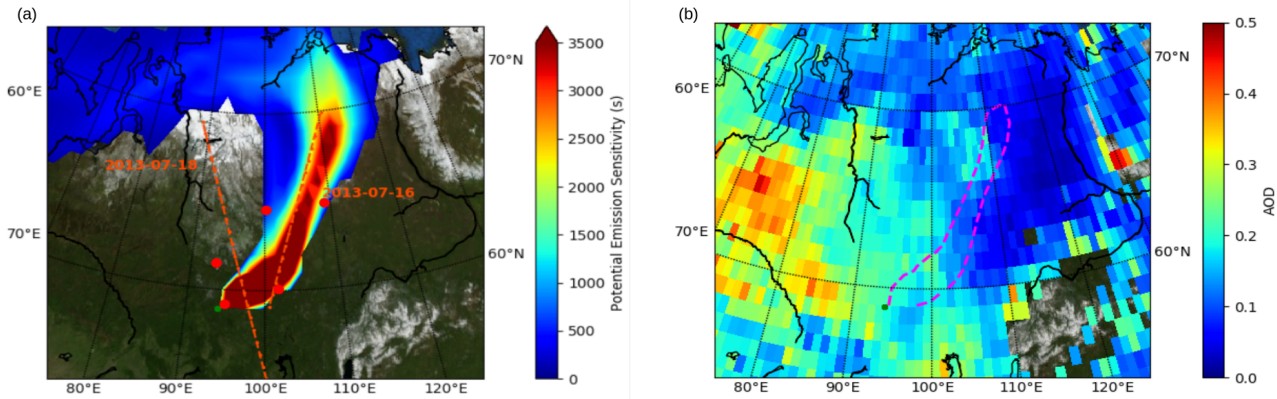

**Figure 10.** (a) : Map of the vertically integrated PES distribution for FLEXPART backward simulation for the aerosol layer at 1.7 km (green point). The red dots represent position of the forest fires detected by MODIS and VIIRS and The orange dotted lines is the selected CALIOP overpasses in the aerosol emission area. (b) : MODIS AOD 1°x1°averaged over the 4 days before the flight (June 16 to 19 2017) with high PES area (PES $\geq$ 2000 s) shown by the pink dotted line. The green point is the airborne lidar position. Map background : NASA's Earth Observatory.

## 4.4 Aged forest fires

It is also interesting to investigate if aged fires correspond to larger lidar ratio as expected according to previous studies (Müller et al., 2005; Tesche et al., 2011; Burton et al., 2013). Part of the June 18, 2017 flight between Novosibirsk and Surgut meets this objective. The latitudinal cross-section PR2, when the aircraft flies at 4 km, shows a layer of aerosol in the altitude range of
0 and 2 km with a horizontal extension of 120 km (Fig. 11). The AOD and LR are calculated for a 9 min average profile shown




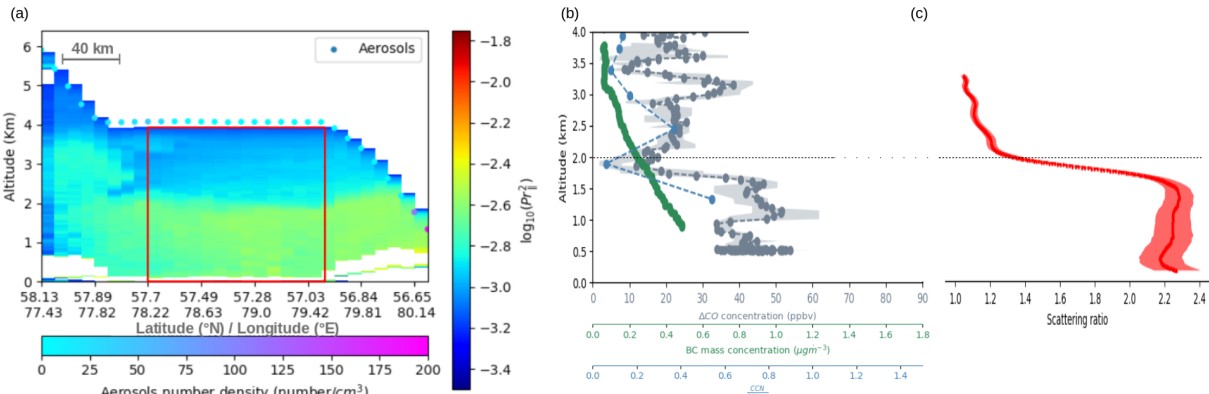

**Figure 11.** (a) : Vertical cross-section of aircraft $\log_{10}(PR^2)$ on June, 18 2017. Calibration constant is $83014 \pm 5\%$. Grimm aerosol concentrations in particle.cm$^{-3}$ are shown at the aircraft altitude. (b) : $\Delta$CO, CCN/Aitken and BC vertical profiles. (c) Aircraft averaged backscatter ratio vertical profile.

by the red rectangle in Fig. 11. The lidar ratio range is between 64 sr and 86 sr when using 0.16 and 0.22 for respectively the $25^{th}$ and $75^{th}$ percentiles of the nearby MODIS 550 nm AOD. The corresponding 532 nm AOD for the airborne lidar is then $0.19 \pm 0.04$ (Table 1), 1.6 times larger than the previous dusty mix case observed at 57°N two days earlier.

The PES calculated by FLEXPART between 500 m and 2 km at 57.4°N (Fig. 12) shows that aerosol emissions are from an area
between 80°E and 100°E at 57°N, while the highest MODIS AOD in this area are related to forest fires which occurred 100°E North of Irkutzk ($0.05 < FRP < 0.2$ GW). The tropospheric CO columns measured by IASI (Fig. 13) also show high values North East of Irkutsk (2.0-2.5 x $10^{18}$ molecule.cm$^{-2}$) while it remains in the range 1.5 - 2.0 x $10^{18}$ molecule.cm$^{-2}$ above the cities of Tomsk, Novosibirsk and Krasnoyarsk. Forest fire plumes from Eastern Siberia can explain the aerosol layers seen by the airborne lidar at 57.49°N, 78.63°E. In-situ aircraft measurements (Fig. 11) are also consistent with this hypothesis since
$\Delta$CO concentrations are even higher than in the fresh fire plume (35-60 ppbv) and the 0.58 CCN to Aitken size distribution ratio is significantly higher than the small values encountered in the fresh fire. The BC mass concentration (0.3 and 0.55 $\mu g.m^{-3}$) is however weaker than the fresh forest value due to the short lifetime (few days) of BC in the atmosphere (Cape et al., 2012; Lund et al., 2018).

### 4.5 Siberian urban and industrial emissions

The aircraft flew over the major Siberian cities in July 2013 between Krasnoyarsk and Novosibirsk, many aerosol layers have been encountered in the 0-3 km altitude range. On example has been selected on July $20^{th}$ 2013 and is shown in Fig. 14. The PR2 latitudinal cross-section when the aircraft is flying at 4.5 km and then descends to 2.5 km, shows a 600 km long aerosol layer in the 0-3 km altitude range. The AOD and LR are calculated for a 1 min average profile shown by the red rectangle in Fig. 14 when the aircraft is between Tomsk and Novosibirsk, i.e. a densely populated area with numerous industrial infrastructure.





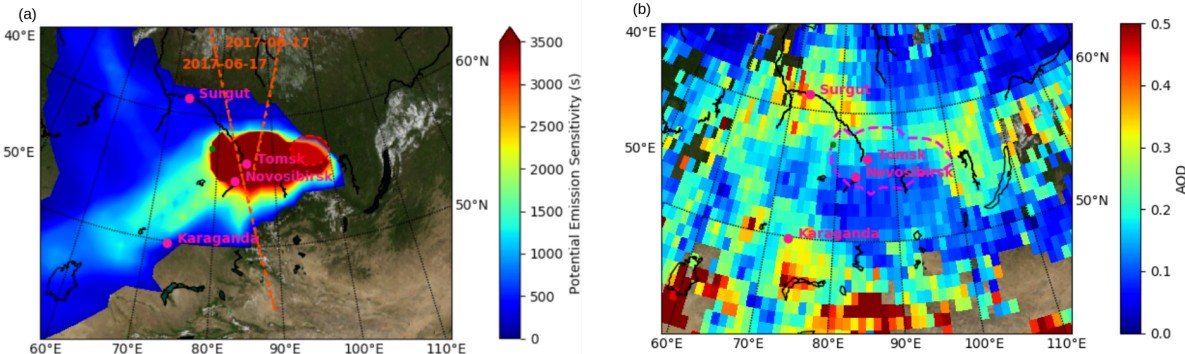

**Figure 12.** (a) : Map of the vertically integrated PES distribution from FLEXPART backward simulation for the aerosol layer between 0-2 km at 57.49°N (green point). The red dotted area is the forest fire area and the orange dotted lines are the selected CALIOP overpasses in the source area. (b) : MODIS AOD 1°x1°averaged over the 4 days before the flight (June 15 to 18 2017) with high PES area (PES $\geq$ 2000 s) shown by the pink dotted line. The green point is the airborne lidar position. Map background : NASA's Earth Observatory.

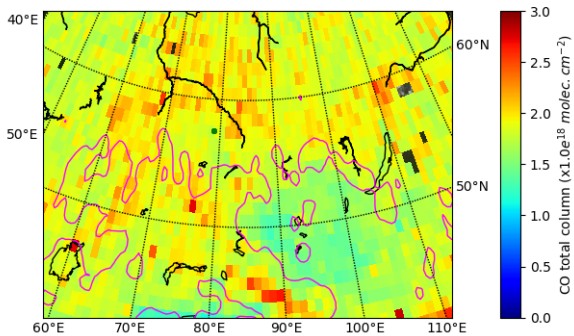

**Figure 13.** Tropospheric CO total column averaged over 4 days before the flight (June 15 to 18 2017). The pink area is the area with dust AOD > 0.08 (AOD at 10$\mu$m from IASI). Map background : NASA's Earth Observatory.

The lidar ratio range is between 71 sr and 90 sr when using 0.13 and 0.21 for respectively the $25^{th}$ and $75^{th}$ percentiles of the nearby MODIS 550 nm AOD (Table 1). The corresponding 532 nm AOD for the airborne lidar is then 0.17 $\pm$ 0.05 (Table 1). The strong PES values obtained with the FLEXPART backward simulation for the aerosol layer observed by the lidar (Fig. 15) remain concentrated between 50°N and 55°N south of Novosibirsk and Tomsk. The range of the 4-day averaged MODIS AODs at 550 nm is 0.2-0.3 in the northern part of this domain near the major cities while it is 0.1-0.2 in the southern part
around 52°N. The tropospheric CO column values above 2.0 x $10^{18}$ molecule.cm$^{-2}$ match also the MODIS AOD distribution (Fig. 16). No fires have been detected in this region using the FIRMS data set and the amount of dust is very low in this area





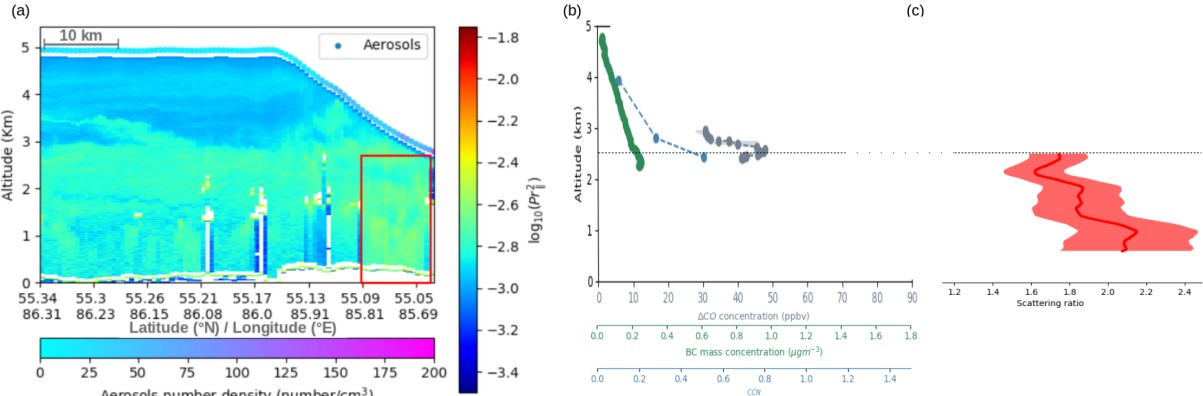

**Figure 14.** (a) : Vertical cross-section of aircraft $\log_{10}(PR^2)$ on July, 20 2013. Calibration constant is $136203 \pm 5\%$. Grimm aerosol concentrations in particle.cm$^{-3}$ are shown at the aircraft altitude. (b) : $\Delta$CO, CCN/Aitken and BC vertical profiles. (c) Aircraft averaged backscatter ratio vertical profile.

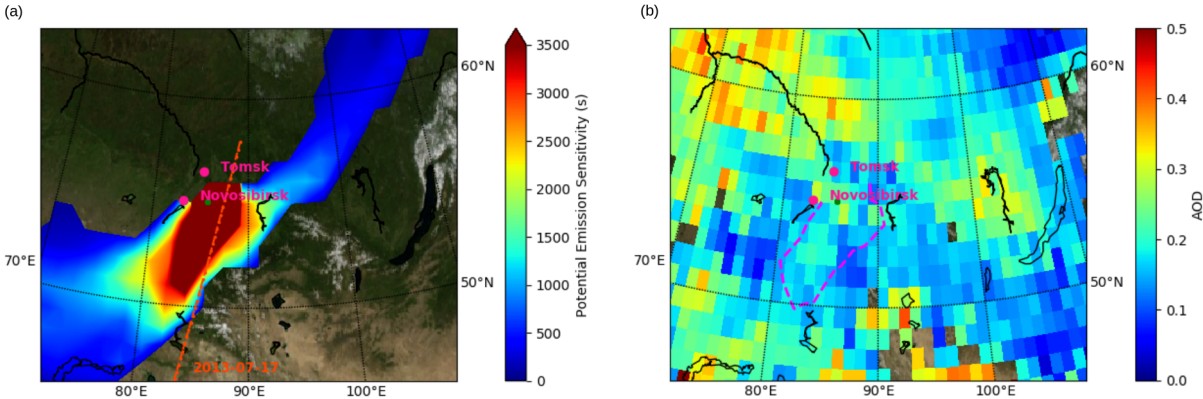

**Figure 15.** (a) : Map of the vertically integrated PES distribution from FLEXPART backward simulation for the aerosol layer between 0-2.5 km at 87.5°N (green point). The orange dotted lines are the selected CALIOP overpasses in the source area. (b) : MODIS AOD 1°x1° averaged over the 4 days before the flight (July 17 to 20 2013) with high PES area (PES $\geq$ 2000 s) shown by the pink dotted line. The green point is the airborne lidar position. Map background : NASA's Earth Observatory.

according to the IASI 10 $\mu m$ AOD which is always below 0.1. Therefore the source of the aerosol plume observed by the airborne lidar is related to the local urban and industrial emissions from the Novossibirsk/Tomsk area. The $\Delta$CO concentration and CCN to Aitken ratio measured by the aircraft at 2 km are respectively 50 ppbv and 0.5 at 2 km altitude, i.e. similar to the values encountered for local urban and industrial emissions in section 4.5. The BC mass concentration (0.2 however lower than the BC values observed for the region dominated by gas flaring emission (0.4-0.5 $\mu g.m^{-3}$).





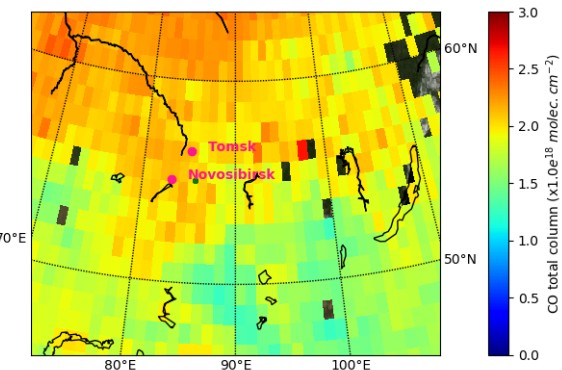

**Figure 16.** Tropospheric CO total column averaged over 4 days before the flight (June 17 to 20 2013). Map background : NASA's Earth Observatory.

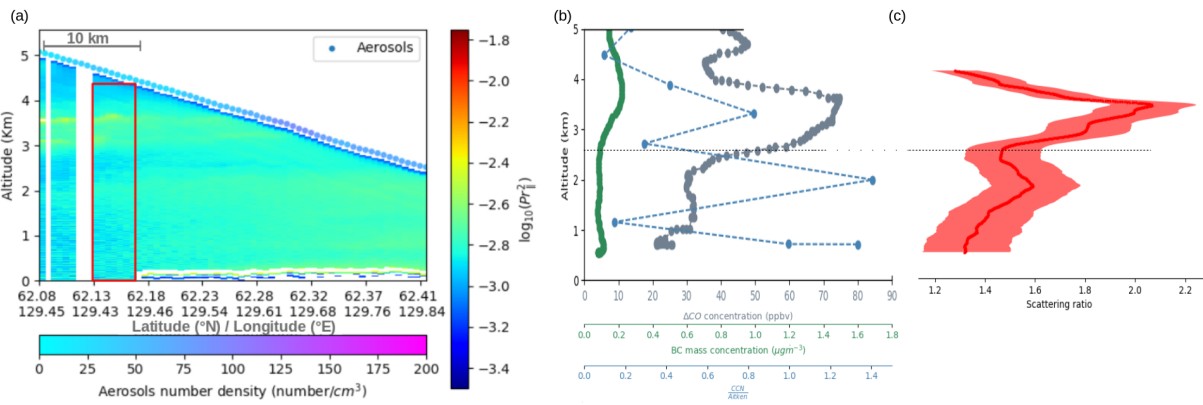

**Figure 17.** (a) : Vertical cross-section of aircraft $\log_{10}(PR^2)$ on July, 19 2013 above Yakutsk. Calibration constant is $127216 \pm 5\%$. Grimm aerosol concentrations in particle.cm$^{-3}$ are shown at the aircraft altitude. (b) : $\Delta$CO, CCN/Aitken and BC vertical profiles. (c) Aircraft averaged backscatter ratio vertical profile.

## 4.6 Long range transport of Northern China emissions

The last case study corresponds to the lidar observations near the city of Yakutzk (62°N, 129°E) on July 19, 2013. A PR2 vertical cross-section when descending to Yakutsk shows several aerosol layers in the 0 and 5 km altitude range (Fig. 17). The

AOD and LR are calculated for a 40 sec average profile shown by the red rectangle in Fig. 17 when the aircraft is high enough to sample the entire aerosol layer between 0 and 4 km. The lidar ratio range is between 41 sr and 51 sr when using 0.1 and 0.13 for respectively the $25^{th}$ and $75^{th}$ percentiles of the nearby MODIS AOD at 550 nm (Table 1). The corresponding AOD for the airborne lidar at 532 nm is then $0.12 \pm 0.04$ (Table 1).





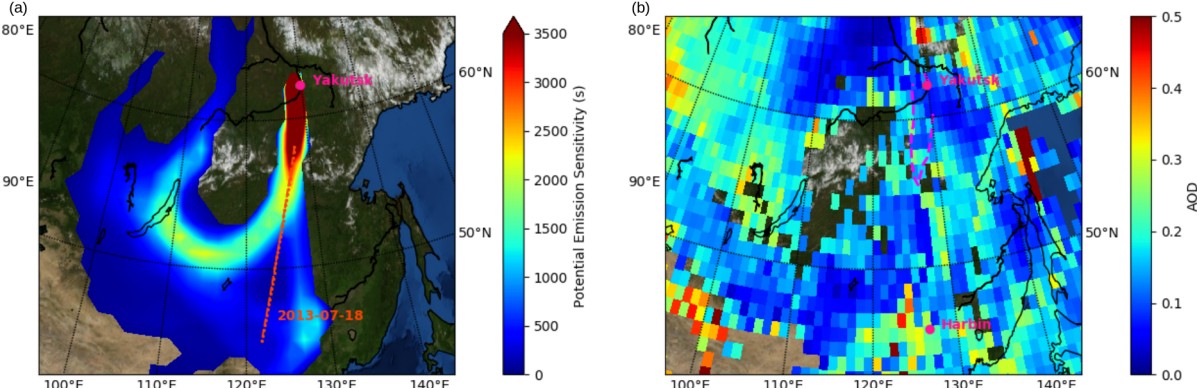

**Figure 18.** (a) : Map of the vertically integrated PES distribution from FLEXPART backward simulation for the aerosol layer between 0-4 km at 62°N, 129.45°E (green points). The orange dotted lines are the selected CALIOP overpasses in the source area. (b) : MODIS AOD 1°x1°averaged over the 4 days before the flight (July 16 to 19 2013) with high PES area (PES ≥ 1000 s) shown by the pink dotted line. The green point is the airborne lidar position. Map background : NASA's Earth Observatory.

The PES calculated by FLEXPART for this aerosol layer and the MODIS AOD 4-day map show that the aerosol source region
is located in a North-South corridor at 128°E extending southward to the city of Harbin in Northern China (Fig. 18). The CO tropospheric column measured by IASI is less than 2.0 x $10^{18}$ molecule.cm$^{-2}$ between Yakutsk and 52°N, while it is higher than 2.0 x $10^{18}$ molecule.cm$^{-2}$ in the Harbin area (Fig. 19). No fire is detected by FIRMS and no dust layer is seen by the 10 $\mu m$ AOD measured by IASI in the area between Harbin and Yaktusk (Fig. 19). Therefore, the main source of the aerosol layers observed by the lidar, and in particular the one located at altitude between 2.5 and 4 km, can be attributed to emissions
from the Harbin region.

The $\Delta$CO concentration measured by the aircraft during an ascent west of Yakutsk at 128.6°E is as large as 75 ppbv in the layer between 2 and 4 km, while the CCN to AN ratio is characteristic of aged aerosol with a value as high as 1.4 (Fig. 17). This is consistent with the hypothesis of atmospheric transport of a polluted aerosol plume between Harbin and Yakutsk. The BC mass concentrations are similar to the values encountered for the previous case when flying around the Siberian cities. So
either the contributions of Harbin and of Novosibirsk/Tomsk emissions to the BC atmospheric concentrations are similar or BC from Harbin emissions has been efficiently removed during the 2-day transport between Harbin and Yakutsk.

### 4.7 Aerosol classification discussion

The characteristics of the 6 aerosol layers studied in this section are grouped in a summary table (Table 1). It can be noted that the highest AODs (> 0.2) were obtained on the one hand for the gas flaring emission in accordance with several studies on the
impact of these emissions on aerosol production (Stohl et al., 2013; Elvidge et al., 2016), and on the other hand for aged fire plumes as frequently observed in Siberia (Damoah et al., 2004; Paris et al., 2009b). However, the highest $\Delta$CO values were obtained for cases of long range transport of combustion aerosols from fires or Northern China. It is consistent with the CO





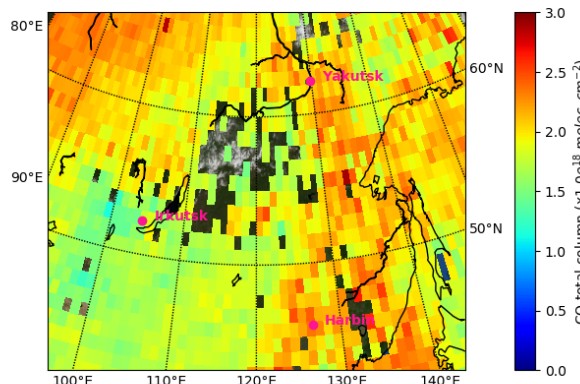

**Figure 19.** Tropospheric CO total column averaged over 4 days before the flight (July 16 to 19 2013). Map background : NASA's Earth Observatory.

lifetime being longer than 15 days and the buildup with time of individual plumes at remote locations (Forster et al., 2001; Petetin et al., 2018). BC mass concentration is high ($> 1.5\ \mu\text{g.m}^{-3}$) for the fresh biomass burning plume while it is less than

$0.5\ \mu\text{g.m}^{-3}$ for the other cases because of efficient BC removal for transport lifetime $> 1\text{-}2$ days (Cape et al., 2012; Lund et al., 2018). Gas flaring contribution to BC mass concentration ($0.5\ \mu\text{g.m}^{-3}$) is also higher than the urban pollution contribution ($0.2\ \mu\text{g.m}^{-3}$).

Regarding the LR values according to the aerosol type, the lowest range is obtained for the "Dusty Mix" case (26 sr - 40 sr) and the highest for the urban pollution from Tomsk and Novosibirsk (71 sr - 90 sr). A lofted dust layer in Tajikistan discussed by

Hofer et al. (2017) with a multiwavelength Raman lidar has LR of the order of 36 sr at 532 nm, while the range of the LR for dusty mix cases discussed by Burton et al. (2013) with an airborne HSRL lidar at 532 nm is between 26 sr and 49 sr in North America. Our dusty mix case shows similar values. The ranges obtained for our fresh and aged biomass burning plumes are also in good agreement with previous studies. Burton et al. (2013) also found fresh biomass burning LR (33-46 sr) lower than aged fire plume LR (55-73 sr) in North America. Lidar ratios reported for aged Siberian smoke layers transported over Japan

(Murayama et al., 2004) or Korea (Noh et al., 2008) are also of the order of 65 sr at 532 nm.

The range of LR values for pollution aerosol is quite large in published papers (43-90 sr) depending on the type of anthropogenic emission, on the season and on the altitude of the aerosol layer. For example in Europe, Chazette et al. (2005) found LR between 60 sr and 77 sr in Paris, while Müller et al. (2007) obtained 45-60 sr in Leipzig. Burton et al. (2013) proposed a LR range between 53 sr and 70 sr in North America using their airborne lidar flights around major North American cities

and Mexico. In Beijing, Xie et al. (2008) found 44 sr for high humidity and heavy pollution and 61 sr for lower humidity and moderate pollution level. Ansmann et al. (2005) and Heese et al. (2017) also found LR generally below 50 sr for the Pearl River delta in China where high pollution and humidity are generally found. Our values for transport of aged aerosol layer from





**Table 1.** Aerosol type / LR classification obtained from the airborne lidar study. For each identified aerosol type, the studied layer AOD, the MODIS AOD distribution used for the inversion and *In-situ* measurements of $\Delta$CO, CCN/Aitken and BC are also presented. Finally, the comparison with Burton et al. (2013) classification is presented.

| Aerosol type | Transport of dusty aerosol mixture from Kazakhstan | Fresh smoke from Siberian fires | Transport of smoke from Siberian fires | Gas flaring industrial emissions | Urban and industrial emissions of Siberian cities | Transport of urban and industrial emissions from Northern China |
|---|---|---|---|---|---|---|
| $\Delta$CO in ppbv | 20 | 35 - 45 | 65 - 60 | 20 - 40 | 50 | 30 - 80 |
| Black carbon concentration ($\mu g.m^{-3}$) | 0.2 - 0.4 | 0.8 - 1.6 | 0.3 - 0.55 | 0.4 - 0.5 | 0.2 | 0.2 |
| CCN / Aitken | - | 0.1 - 0.15 | 0.6 | 0.3 - 0.65 | 0.5 | 0.8 - 1.4 |
| Aircraft AOD | $0.0895 \pm 0.0225$ | $0.117 \pm 0.02$ | $0.19 \pm 0.04$ | $0.23 \pm 0.06$ | $0.17 \pm 0.05$ | $0.12 \pm 0.04$ |
| Aircraft LR | 26 - 40 | 34 - 40 | 64 - 86 | 43 - 60 | 71 - 90 | 41 - 51 |
| MODIS AOD ($25^{th}$ - $75^{th}$) | 0.069 - 0.112 | 0.115 - 0.125 | 0.16 - 0.22 | 0.2 - 0.26 | 0.13 - 0.21 | 0.1 - 0.13 |
| Aerosol type Burton et al. (2013) classification | Dusty Mix | Fresh Smoke | Smoke | Urban | Urban | Polluted Marine |
| LR ($5^{th}$ - $95^{th}$) Burton et al. (2013) classification | 29 - 49 (14 - 63) | 33 - 46 (24 - 52) | 55 - 73 (46 - 87) | 53 - 70 (43 - 81) | 53 - 70 (43 - 81) | 36 - 45 (27 - 50) |

China is in the lower range (40-50 sr) as expected for aged aerosol and southerly flow with many clouds and high humidity over Eastern Siberia (Fig. 18). The "polluted marine" aerosol type in the Burton et al. (2013) classification better corresponds

to the aged aerosol layer from Northern China, as their aircraft flights are then representative of aged anthropogenic plumes from coastal cities transported over the ocean. The LR range of 36-45 sr is then similar to this work for the aged plume from Northern China.

Our LR values for the Novosibirsk area (70-90 sr) is in the upper range of previous observations of polluted aerosol layers. Dieudonné et al. (2017) also found elevated LR (> 90 sr) at 355 nm around several cities in Russia. Since the lidar ratio is

generally 10 sr higher at 355 nm than at 532 nm (Mattis et al., 2004; Müller et al., 2007), these results are consistent with the high LR values found by the airborne lidar around Novosibirsk.





Finally the LR range obtained for the gas flaring area (43 sr - 60 sr) cannot be compared to an existing value as specific studies of these aerosol plumes with a lidar system have not been published. Chazette et al. (2018) reported LR of the order of 71 sr at 355 nm in Hammerfest, Northern Norway for a mixture of flaring emissions from a local industrial source and transport of
pollution aerosol from the Murmansk area in Russia and is hardly comparable to the case of the Ob Valley with many flaring sources. The range of LR (40-50 sr) measured in Tomsk above the boundary layer between 2.5 and 5 km using a Raman lidar at 532 nm in April-May 2007 (Samoilova et al., 2010) is probably a better proxy of the influence of the Ob Valley flaring emission area since no fires are present over Siberia during this month and long range transport is mainly controlled by Westerly and Northerly wind at 500 hPa from the Ob Valley region (22 days out of 31).

## 5   Comparison of airborne lidar and CALIOP aerosol layer data products

In this section, aerosol type and LR values derived from the airborne lidar data analysis are compared to nearby CALIOP overpasses. The time of passage and the position of the aircraft do not allow to have an exact coincidence with the footprint of the CALIOP lidar. Therefore CALIOP granules to be compared with the aircraft have been selected using the FLEXPART PES maps discussed in the previous section. For the selected CALIPSO granule, cloud free attenuated backscatter profiles with PES
$> 0.8 \cdot \text{PES}_{\text{max}}$ are averaged to obtain a mean backscatter ratio vertical profile using the methodology described in Ancellet et al. (2014). The averaging distance is generally of the order of 100-300 km (see Table 2). The aerosol type and the range of the LR distribution for the CALIOP profile is then taken from the Version 4.1 level 2 CALIOP aerosol data product Kim et al. (2018).

The comparisons of the total backscatter ratio vertical profiles for the six case studies presented in Table 1 are shown in Fig.
20. The positions and dates of the CALIPSO profiles chosen for this comparison are given in Table 2 along with the spatial and temporal differences between the aircraft measurements and the CALIOP profiles. Only one case above the Ob valley on June 18 (Fig. 20d) corresponds to a distance less than 150 km and a time lag less than 12 h. For the other cases, a strong sensitivity to a spatial distance larger than 200 km is only expected on June 16 (Fig. 20a) when the air mass transport direction is not parallel to the line connected the aircraft and CALIOP profile positions, and when the differential advection of the aerosol
plume may change the vertical structure of the total backscatter. The range of the selected CALIOP backscatter ratio profiles in Fig. 20 differs from that of airborne lidar by less than 20% and the thicknesses of the aerosol layers are in good agreement. The comparison for the case with long range transport from Northern China even shows a surprisingly good agreement (Fig. 20f) considering the large distance (576 km) between the two measurements mainly because the transport pathway is parallel to the line connected the CALIOP footprint and the aircraft position. Because the sensitivity to spatial and temporal mismatches is
expected to be large for the biomass burning cases, two CALIOP vertical profiles are selected in Fig. 20b, c provided that the $0.8 \cdot \text{PES}_{\text{max}}$ criteria is still true. For the fresh fire case (Fig. 20b), a layer with similar structure and backscatter ratio magnitude was seen by CALIOP 60 hours earlier and 500 km further north, and should be also considered in the comparison. For the aged forest fires, the differences between the two CALIOP vertical profiles are quite small and the CALIOP profiles are then fairly representative of the airborne lidar observations. The 532 nm AOD calculated for the aircraft and CALIOP profiles (Table





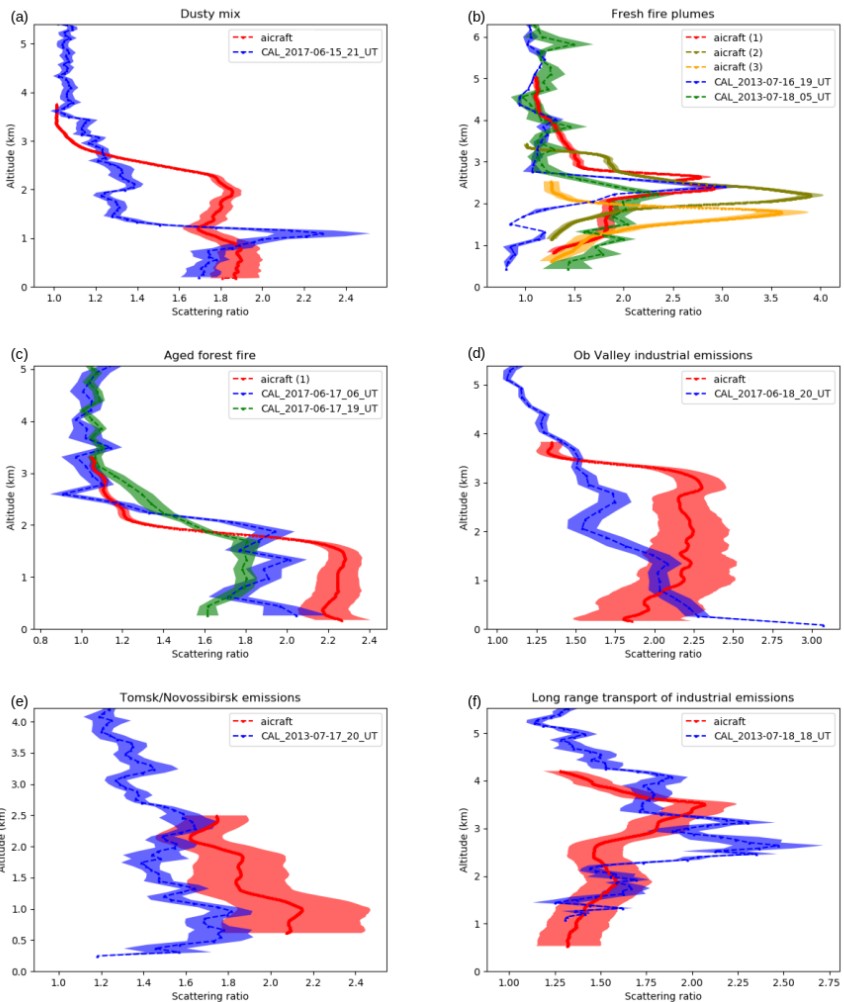

**Figure 20.** Comparison of aircraft and CALIOP averaged backscatter ratio vertical profile.

2) are very similar except for the aged urban pollution layers transported from northern China where the AOD differences between CALIOP and the airborne lidar is 0.16 (+57%). This is mainly due to large difference of lidar ratio (70 sr instead 45 sr) between the CALIOP retrieval and the aircraft data (see hereafter). Therefore the selected CALIOP profiles are suitable to discuss the differences between the CALIOP V4.2 lidar ratio values compared to the airborne lidar data analysis.

The aerosol composition for the six averaged CALIOP profiles are given in Table 2 using the CALIOP level 2 aerosol type and taking into account the thickness of the aerosol layers. The corresponding lidar ratio is obtained using the fraction of the aerosol type and the lidar ratio range given for each type of the CALIOP aerosol classification by Kim et al. (2018); Omar et al. (2009). For the forest fire cases where two CALIOP profiles have been considered for the comparison, the two CALIOP profiles have the same composition for the fresh fire case and the mixture of polluted dust and smoke is only slightly different





for the two CALIOP profiles selected for the analysis of the aged forest fire plume (less polluted dust fraction for the CALIOP

profile least distant from the Baikal lake fire area).

Regarding the aerosol type found for the CALIOP observations, there is a good agreement with our detailed analysis of the aerosol sources for three cases (dusty mixture, fresh smoke and aged pollution plume from Northern China) while the classification is not appropriate for the aged Siberian fires from Lake Baikal region and for the urban and gas flaring emissions from Russia. For the aged forest fire case and the gas flaring emissions, the fraction of polluted dust aerosol is too high. These

two misclassifications led to moderate underestimates of the lidar ratio of 5 sr and 7 sr respectively for the case of old fires and flaring gas emissions. In fact the misclassification even compensates for the bias on the value of the lidar ratio of the polluted continental type which is too high for gas flaring emissions. The misclassification of the CALIOP profile related to urban and industrial Russian emissions seen as an elevated smoke type occure mainly because the polluted PBL thickness is frequently higher than 2.5 km in summer above Siberia (i.e. the upper limit to ascribe an elevated smoke type to an aerosol layer in the

CALIOP V.4). This misclassification does not impact the value of the lidar ratio.

Eventhough the aerosol type classification is correct for the dusty mixture and the transport of pollution plume from Northern China, the lidar ratio found for CALIOP is too high by 25 sr. Although this difference is not far from the CALIOP lidar ratio uncertainty, it is likely that aerosol plume aging and mixture with background aerosol cannot be properly taken into account and lead to a positive bias when deriving the lidar ratio from the aerosol type. Regarding the fresh forest fire case, CALIOP

classification is correct but since the age of the fire is not taken into account in the CALIOP data processing the lidar ratio is two times larger than the estimated value by the airborne lidar. This will lead to an overestimation of the aerosol AOD by CALIOP when sampling biomass burning plume very close to the fire region.

## 6   Conclusions

Two airborne lidar campaigns were carried out over Siberia in July 2013 and June 2017. Aerosol types and optical properties

were derived using FLEXPART, satellite data and airborne *in-situ* measurements when available. Six aerosol type could be identified in this work: (i) Dusty aerosol mixture (ii) Ob valley industrial emission (iii) fresh boreal forest fire plumes (iv) aged forest fire plumes (v) pollution over the Tomsk/Novosibirsk region (vi) long range transport of Chinese pollution over Yakutsk. The aircraft in-situ measurement, mainly $\Delta$CO and BC have been useful to validate the identification of the aerosol origin using FLEXPART and the satellite observations ; namely large BC concentrations in the fresh forest fire plume and large $\Delta$CO

for the long range transport of Eastern Siberian forest fires and of polluted plumes from Northern China. The lidar ratio (LR) analysis shows that the lowest LR range is obtained for the "Dusty Mix" case (26-40 sr) and the highest for the urban and industrial pollution from the Tomsk/Novosibirsk area (71-90 sr). We found a good agreement of this work analysis of the LR values according to the aerosol classification with previous studies (e.g., Burton et al., 2013). The range of lidar ratio obtained for gas flaring emission (43-60 sr) is lower than the high values encountered in the Tomsk/Novosibirk urban area and has never

been characterized using lidar observations.





**Table 2.** Aerosol type / LR classification and AOD obtained from the study of airborne lidar data and the associated CALIOP profiles are presented. Spatial and temporal informations of CALIOP profiles are also presented.

| Aerosol type | Transport of dusty aerosol mixture from Kazakhstan | Fresh smoke from Siberian fires | Transport of smoke from Siberian fires | Gas flaring industrial emissions | Urban and industrial emissions of Siberian cities | Transport of urban and industrial emissions from Northern China |
|---|---|---|---|---|---|---|
| CALIOP time | 15/06/2017, 21UT | 16/07/2013, 19UT<br>18/07/2013, 05UT | 17/06/2017, 06UT<br>17/06/2017, 19UT | 18/06/2017, 20UT | 17/07/2013, 20UT | 18/07/2013, 18UT |
| CALIOP HAV (km) | 190 | 60<br>100 | 220<br>220 | 360 | 140 | 110 |
| CALIOP mean position | 54.5°N,82.5°E | 61.8°N,102.4°E<br>59.3°N,97.6°E | 56.7°N,83.7°E<br>57.3°N,87°E | 57.85°N,76.62°E | 55.2°N,87.52°E | 57°N,127.15°E |
| $D_x$ (km) | ≈ 200 | ≈ 530<br>≈ 220 | ≈ 375<br>≈ 525 | ≈ 224 | ≈ 105 | ≈ 640 |
| $\Delta t$ (h) | -7 | -60<br>-24 | -20<br>-7 | +19 | -57 | -12 |
| CALIOP aerosol type (LR of aerosol type): | | | | | | |
| Polluted Dust (55 ± 22 sr) | 84% | 11 % | 29%<br>39% | 65% | 10% | 0% |
| Polluted continental / Smoke (70 ± 25 sr) | 16 % | 89% | 71%<br>56% | 32% | 0% | 70% |
| Elevated smoke (70 ± 16 sr) | 0 % | 0 % | 0%<br>0% | 0% | 90% | 30% |
| Clean continental (53 ± 24 sr) | 0 % | 0 % | 0 %<br>5% | 3% | 0% | 0% |
| CALIOP mean LR | 57±23 | 70±16 | 66±25<br>63±25 | 60±24 | 69±25 | 70±25 |
| CALIOP AOD | 0.093 ± 0.04 | 0.09 ± 0.04<br>0.18 ± 0.07 | 0.19 ± 0.07<br>0.14 ± 0.06 | 0.19 ± 0.06 | 0.11 ± 0.03 | 0.28 ± 0.1 |
| Aircraft LR | 26 - 40 | 34 - 40 | 64 - 86 | 43 - 60 | 71 - 90 | 41 - 51 |
| Aircraft AOD | 0.089 ± 0.02 | 0.2 ± 0.02 | 0.19 ± 0.04 | 0.23 ± 0.06 | 0.17 ± 0.05 | 0.12 ± 0.04 |



Airborne lidar backscatter ratio vertical structure, aerosol types and integrated LR derived from the airborne data analysis (section 4) were compared to nearby CALIOP overpasses. We found three main differences with the CALIOP LR and aerosol type classification over Siberia: (i) layer can be classified as Elevated smoke instead of Polluted continental and vice versa, but with little influence on the LR value (ii) aging and transport of aerosol layers effect on the LR value is not always properly

accounted for even when the classification is correct (e.g. the dusty mixture is properly identified but with a lidar ratio too high) (iii) the lack of discrimination between fresh and old fire plume leads to an overestimation of the optical depth for the fresh fires. Constrained LR CALIOP with an independent AOD could be another alternative to alleviate some of these limitations discussed in this paper. Such an independent AOD value could be given by co-localized MODIS observations especially for daytime observations in summer. Surface lidar reflectance observations on homogeneous surfaces such as the Siberian taiga or

artic tundras could be also a very good alternative as discussed by (Josset et al., 2018) especially for nighttime observations.



*Code availability.* The FLEXPART code version 9.2 was downloaded from the FLEXPART wiki homepage (https://www.flexpart.eu/downloads).

*Data availability.* Airborne lidar data and *in-situ* measurements are available and can be provided on request (contact : Mikhail Arshinov -
michael@iao.ru). The daily MODIS and VIIRS information from the fires were provided by LANCE FIRMS operated by NASA/GSFC/EOSDIS
and are available at https://firms.modaps.eosdis.nasa.gov/download/. MODIS AOD product (10 km and gridded 1°resolution) were down-
loaded in hdf format at https://ladsweb.modaps.eosdis.nasa.gov/archive/. The 10 $\mu m$ IASI product was download in NetCDF format at
https://ara.lmd.polytechnique.fr/index.php?page=aerosols. CALIOP level L2 data have been downloaded from the ICARE date base (http:
//www.icare.univ-lille1.fr). The AERIS infrastructure (http://www.aeris-data.fr) provided the access to the IASI CO data. Meteorological
Analysis are available at ECMWF (http://www.ecmwf.int)

*Competing interests.* No competing interests are present

*Acknowledgements.* The work was supported by Sorbonne Université and the French Centre National d'Etudes Spatiales (PhD contract
n°2921/2017). The work was also supported in part by Ministry of Science of Education of RF (Agreement No. 05.616.21.0118, unique
identifier RFMEFI61619X0118). We thank the European Centre for Medium Range Weather Forecasts (ECMWF) for the provision of ERA-
Interim reanalysis data and the FLEXPART development team for the provision of the FLEXPART 9.2 model version used in this publication.
The authors thank the AERIS infrastructure and NASA/GSFC for providing the satellite data used in this paper (CO, AOD, CALIOP and fire
FRP).



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
