# Peer review of "Investigating emission sources and transport of aerosols in Siberia using airborne and spaceborne LIDAR measurements"

_Atmospheric Chemistry and Physics, 2020_

## Referee Comment (RC1) · Anonymous Referee #2 · 19 May 2020

This paper aims contributing to improve the scientific knowledge about aerosol particles over Siberia, a region where some aerosol types (such as pollutants) are still understudied. Because aerosol profiling measurements are scarce in this part of the Earth, this study proposes a combination of in-situ, lidar airborne and satellite-based measurement in the framework of two campaigns to reduce the lack of atmospheric information.

In general, this paper contributes to improve the scientific knowledge but its scientific sound would increase much more if the database were enlarged in the future, with a dataset capable of being statistically analyzed. Anyway, I consider this work (with

limited impact) might be published in Atmospheric Chemistry and Physics after major revisions.

Specific comments:

Line 57: This statement is far from truth. Surface measurements are very valuable from a local point of view (with high vertical and temporal resolution). It is not fair to say that they are limited to a few case studies when there are lidar networks that have been working for up to 20 years in some cases. A comment on the main lidar monitoring networks (EARLINET, MPLNET, LALINET, Cislinet, ADnet, NDACC) should be included.

Lines 99-100: Which method was used to perform the polarization calibration? Polarization calibration seems to be not enough for aerosol typing. Can you explain in more detail this? If so, taking into account that co and cross-polarized signal are separately detected, is the optical combination of both signals quality assured?

Lines 101-102: This threshold seems to be somehow ambiguous. Can you provide some references? Is this value cloud type dependent?

Lines 115-116: The correction factor applied for deriving the black carbon mass concentration hugely ranges from 0.5 to 1, depending on the blackening. Because of the impact that this correction (up to 100%) has on the final product, this issue should be explain more in detail.

Line 126: Which is the corresponding accuracy for wind vector?

Line 134-135: For overlap characterization, a method based on the ratio between PR2 and molecular backscatter profile is employed. The latter is computed from the ERA Interim ECWF meteorological analysis, which can differ slightly from the actual atmospheric temperature and pressure values. Is this uncertainty account for the calculations? Which is the impact of using ERA Interim in the overlap derivation? How much does the overlap change from flight to flight?

Line 139-142: The constraining of lidar ratio must be carefully performed. One limitation is the spatial distribution because of the highly aerosol variability. How can the authors justify that an area of +-70 km2 does not disturb the actual AOD value? Another limitation is the temporal distribution. Considering a time slot of +5 around the aircraft observation implies roughly 50% of daytime in June/July for your locations. How can you justify steady AOD values for such huge time slot? In addition, MODIS and airborne lidar work at different wavelengths. How do you deal with this?

Line 186: In the different subsections of section 4, I miss a more complete comparison/discussion of the lidar ratio values obtained here respect to those in the previous literature. There are tens of papers in the framework of EARLINET reporting lidar ratio values for many aerosol types. Your discussion would enrich reviewing the main studies.

Lines 323-330: Again, this comparison will enrich including some works performed in the framework of EARLINET. For instance, regarding biomass burning might consider these articles (and references herein):

Baars, H., Ansmann, A., Ohneiser, K., Haarig, M., Engelmann, R., Althausen, D., Hanssen, I., Gausa, M., Pietruczuk, A., Szkop, A., Stachlewska, I.S., Wang, D., Reichardt, J., Skupin, A., Mattis, I., Trickl, T., Vogelmann, H., Navas-Guzmán, F., Haefele, A., Acheson, K., Ruth, A.A., Tatarov, B., Müller, D., Hu, Q., Podvin, T., Goloub, P., Veselovskii, I., Pietras, C., Haeffelin, M., Fréville, P., Sicard, M., Comerón, A., García, A.J.F., Menéndez, F.M., Córdoba-Jabonero, C., Guerrero-Rascado, J.L., Alados-Arboledas, L., Bortoli, D., Costa, M.J., Dionisi, D., Liberti, G.L., Wang, X., Sannino, A., Papagiannopoulos, N., Boselli, A., Mona, L., D'Amico, G., Romano, S., Perrone, M.R., Belegante, L., Nicolae, D., Grigorov, I., Gialitaki, A., Amiridis, V., Soupiona, O., Papayannis, A., Mamouri, R.-E., Nisantzi, A., Heese, B., Hofer, J., Schechner, Y.Y., Wandinger, U., Pappalardo, G. The unprecedented 2017-2018 stratospheric smoke event: Decay phase and aerosol properties observed with the EARLINET (2019) Atmospheric Chemistry and Physics, 19 (23), pp. 15183-15198. DOI: 10.5194/acp-19-

15183-2019

Ortiz-Amezcua, P., Luis Guerrero-Rascado, J., Granados-Munõz, M.J., Benavent-Oltra, J.A., Böckmann, C., Samaras, S., Stachlewska, I.S., Janicka, L., Baars, H., Bohlmann, S., Alados-Arboledas, L. Microphysical characterization of long-range transported biomass burning particles from North America at three EARLINET stations (2017) Atmospheric Chemistry and Physics, 17 (9), pp. 5931-5946. DOI: 10.5194/acp-17-5931-2017

Sicard, M., Granados-Muñoz, M.J., Alados-Arboledas, L., Barragán, R., Bedoya-Velásquez, A.E., Benavent-Oltra, J.A., Bortoli, D., Comerón, A., Córdoba-Jabonero, C., Costa, M.J., del Águila, A., Fernández, A.J., Guerrero-Rascado, J.L., Jorba, O., Molero, F., Muñoz-Porcar, C., Ortiz-Amezcua, P., Papagiannopoulos, N., Potes, M., Pujadas, M., Rocadenbosch, F., Rodríguez-Gómez, A., Román, R., Salgado, R., Salgueiro, V., Sola, Y., Yela, M. Ground/space, passive/active remote sensing observations coupled with particle dispersion modelling to understand the inter-continental transport of wildfire smoke plumes (2019) Remote Sensing of Environment, 232, art. no. 111294, DOI: 10.1016/j.rse.2019.111294

Lines 344-345: Specify that this is only valid for this aerosol type. For others, such as mineral dust this affirmation is not correct.

Line 355 (section 5): Here is my main concern. I do not agree with the scope of this section. Taking into account the datasets you are comparing, this does not make sense at all (huge distance/time for comparison). What might be interesting is the use of CALIOP for complementing the profiling done by the flights and check coherence among datasets, but not comparison. Therefore, please reorganize section 5 in this sense.

Technical comments:

General technical comment: Due AOD is a spectral quantity, it is mandatory to always

identify the wavelength.

Line 32: Paris et al., 2009b should be Paris et al., 2009a.

Line 45: replace 'angstrom' by 'Angström'.

Lines 84-85: insert comma before (ii) and (iii).

Figure 1: include 'Altitude (km asl)' in panels (b) and (c).

Line 95: replace 'mrd' by 'mrad'.

Line 103: replace 'clearing' by 'screening'.

Figure 3, Figure 6, Figure 9, Figure 11, Figure 14 and Figure 17: It is advisable to show the same altitude scale in (a), (b) and (c). In addition, the meaning of the horizontal line in panels (b) and (c) should be included.

Figure 4, Figure 7, Figure 10, Figure 12, Figure 15, Figure 18: Lines for selected CALIOP overpasses and high PES area are somehow difficult to see. I recommend another type of visualization to increase contrast (i.e. white color). Similarly for pink lines in panel (b).

Figure 5, Figure 8: Specify that this is particle linear depolarization ratio, and altitude is in km asl. Line 295: replace '40 sec' by '40 s'.

Table 1: wavelength for AOD and units for lidar ratio are missing.

---

## Referee Comment (RC2) · Anonymous Referee #1 · 20 May 2020

Siberia covers a large area in Asia and little is known about the aerosols above Siberia. Siberian forest fire plumes are transported over large distances. Therefore, it is important to study the optical properties of smoke close to the source and after long-range transport and as well industrial pollution. Airborne lidar measurements offer the great possibility to study various scenarios with different aerosol types. However, the methods used in this paper do not reflect the state-of-the-art approaches and the contribution to the scientific advance is little. Therefore, I have to reject the current version of the manuscript and encourage the authors to stress other results from their campaign.

The reasons for the rejection are the following:

1. The extinction-to-backscatter ratio (lidar ratio) of different aerosol types is one of the main products of this study. However, it is not measured directly, but retrieved using the backscatter coefficient from the airborne lidar and the aerosol optical depth (AOD) from the satellite (MODIS). State-of-the-art would be to use a high spectral resolution lidar (HSRL) which performs direct measurements of the aerosol extinction coefficient profile and can be flown on an aircraft. A ground-based Raman lidar offers another state-of-the-art direct measurement of the extinction coefficient and provides vertical profiles of the lidar ratio. Even a ground-based backscatter lidar with a collocated sun photometer to constrain the extinction coefficient using the AOD would offer more accurate results. Therefore, the scientific advance by this indirect retrieval of the lidar ratio is little. And as the lidar ratio is not just a side product but the main product of the aerosol characterization in this study, it is not sufficiently well constrained.

2. The comparison of the airborne observations with the spaceborne CALIOP measurements is highly uncertain. The temporal and spatial distance is too large to draw valid conclusions. An aerosol plume 500 km away contains not necessarily the same aerosol. In the case of the shortest distance (105 km) the satellite passed more than two days (57 hours) prior to the aircraft over the area. In the last 14 years several much better comparisons have been presented. There are studies using ground-based lidar systems with a spatial distance < 100 km or airborne lidar systems flown along the track of CALIPSO. The flight patterns of the campaigns are not designed for a good comparison with CALIPSO. Therefore, the conclusions drawn with respect to CALIOP might be correct but are not based on a convincing comparison.

3. Additionally, a state-of-the-art aerosol characterization includes a proper measurement of the depolarization ratio (see Burton et al., 2012, or Groß et al., 2013). The depolarization ratio conveys important information about the particle shape and is therefore a key parameter in the aerosol classification presented in Burton et al., 2013, to which the authors refer in the manuscript. I wonder how the authors assign one of Burton's aerosol types in Tab. 1 to their observations by using just one out of the four

parameters used to separate different aerosol types. This is a very rough estimate and does not represent the state-of-the-art.

4. Two out of the six cases presented have significant ambiguities in the current version of the manuscript. Case 2 (Ob Valley gas flaring emissions) is a multilayer scenario. In Fig. 8b a CALIOP measurement is presented showing a dust layer above the Ob Valley emissions. The airborne lidar measurement does not capture this layer and it is not clear, if the dust layer is present above the aircraft. In that case the MODIS AOD is biased by this second layer and can not constrain the lidar ratio of the backscatter lidar. For case 6 (Long-range transport of Northern China emissions) the source appointment is not very convincing. The FLEXPART backward simulation (Fig. 18a) shows a large residence time in the area south of Yakutsk. It is possible that the aerosol originates from Harbin in Northern China, but Fig. 18a suggests an origin between lake Baikal and Yakutsk with a potential emission sensitivity of around 2000 s (shown in yellow). This is not discussed at all. The second ambiguity in this case is the top height of these aerosol layers. From the manuscript it can not be decided, if there are more aerosol layers above 5 km height. In the case of further layers above the aircraft, the extinction can not be constrained by the MODIS AOD.

Minor remarks:

i. Please use always the year when writing the date to avoid ambiguities (especially in Sec. 5).

ii. Comment on the speed of the aircraft and the translation from the temporal resolution of the lidar measurements to a horizontal resolution of the measurements.

iii. Looking at Fig. 4, 7, 10, 15, 18 the pink line in Fig. (b) shows less than the PES > 2000 s, according to the color scale in Fig. (a), especially in Fig. 18 where it is supposed to show PES > 1000 s.

iv. L344/345 "Since the lidar ratio is generally 10 sr higher at 355 nm than at 532 nm"

This is simply not true.

It is a challenging task to characterize well the different aerosol types without direct measurements of the lidar ratio (HSRL) or the depolarization ratio. However, Siberia is an interesting region and the effort put into this work should be appreciated. The method to determine the lidar ratios and the CALIOP comparison is not convincing and not state-of-the-art. However, I would encourage the authors to focus on different aspects of their campaign: The differences between fresh and aged forest fire smoke or between gas flaring emissions and urban pollution. Also, a stronger focus could be put on the comparison between the lidar and the in situ measurements.

---

## Author Comment (AC1) · 15 Jul 2020

We decided to withdraw the current manuscript in order to refocus the results of the airborne campaign towards the analysis of the Siberian aerosol sources and to drop the discussion on the lidar ratio characterization and comparison with CALIOP. Satellite data will be only provided as a mean to document the regional context of the airborne observations.

Some quick answers to the reviewer comments are nevertheless provided on suplementary document.

[Figure]

Please also note the supplement to this comment:
https://www.atmos-chem-phys-discuss.net/acp-2020-195/acp-2020-195-AC1-supplement.pdf

**Supplement:**

Answer Reviewer 1

We decided to withdraw the current manuscript in order to refocus the results of the airborne campaign towards the analysis of the Siberian aerosol sources and to drop the discussion on the lidar ratio characterization and comparison with CALIOP. Satellite data will be only provided as a mean to document the regional context of the airborne observations.

Some quick answers to the reviewer comments are nevertheless provided :

*Line 57: A comment on the main lidar monitoring networks (EARLINET, MPLNET, LALINET, Cislinet, ADnet, NDACC) should be included.*
Yes we agree that the contribution of the ground based lidar networks is not sufficiently recognised. The unfortunate remark referred mainly to the limited amount of measurements available in Russia. This point will be corrected in the new manuscript.

*Lines 99-100: Which method was used to perform the polarization calibration? Polarization calibration seems to be not enough for aerosol typing. Can you explain in more detail this?*
As mentioned in the first version of the paper, the airborne lidar does not have a reliable measurement of the depolarization channel and only a rough estimate of the presence of depolarization can be used. It is used as an additionnal information in the analysis of the intensity of the attenuated backscatter coefficient to filter out the data within cloud layers.

*Lines 115-116: The correction factor applied for deriving the black carbon mass concentration hugely ranges from 0.5 to 1, depending on the blackening. Because of the impact that this correction (up to 100%) has on the final product, this issue should be explain more in detail.*
This issue will be addressed in the new version. Absolute calibration of the aethalometer was performed in the laboratory conditions by means of a pyrolysis generator of soot particles and comparison of the data of simultaneous optical and gravimetric measurements.

*Line 134-135: For overlap characterization, a method based on the ratio between PR2 and molecular backscatter profile is employed. The latter is computed from the ERA Interim ECWF meteorological analysis, which can differ slightly from the actual atmospheric temperature and pressure values. Is this uncertainty account for the calculations? Which is the impact of using ERA Interim in the overlap derivation? How much does the overlap change from flight to flight?*
The error related to the use of the ECMWF analysis for the retrieval of the molecular density vertical profile is quite small (< 1-2%) compared to the assumption of clear sky conditions (5 % error). The shape of the overlap function correction does not change significantly from flight to flight, but the absolute lidar calibration is certainly different from flight to flight.

*Line 139-142: The constraining of lidar ratio must be carefully performed. One limitation is the spatial distribution because of the highly aerosol variability.*
Yes we agree it is a difficult task to address this question without a direct measurement of the extinction profile. We have estimated the uncertainty in the retrieval when using a constraint with MODIS AOD by using the distribution of several MODIS AOD in the air masses sampled by the aircraft. However it is true that the spatio-temporal differences between the satellite observations and the aircraft observations is always a critical question, especially when the aerosol optical properties rapidly change with mixing or relative humidity evolution. The time difference between the MODIS observations and the aircraft observations are less than 1 hour for three cases (dusty mix, fresh fire and Siberian city emissions), while it is indeed between 4-5 h for the other three cases (aged fire, gas flaring, aged urban emissions from China). In the new version of the paper the retrieval of an aircraft extinction profile based on a constraint by MODIS will be left out. The retrieval of an AOD below the aircraft will be made using our analysis of the type of aerosol based

on the FLEXPART analysis coupled with the in-situ measurements, while the lidar ratio for the corresponding aerosol type will be taken according to the existing values of the scientific literature. The MODIS AOD from Aqua and Terra will be only an additionnal information in the analysis of the airborne lidar data to provide the regional context of the aerosol distribution.

*Line 186: In the different subsections of section 4, I miss a more complete comparison/discussion of the lidar ratio values obtained here respect to those in the previous literature.*

It is done in section 4.7 line 323-352 when discussing the airborne lidar AOD dat. With the new approach chosen above, an analysis of the lidar ratios used in the litterature will be carried out prior to the analysis of the airborne lidar measurements.

*Line 355 (section 5): Here is my main concern. I do not agree with the scope of this section. Taking into account the datasets you are comparing, this does not make sense at all (huge distance/time for comparison). What might be interesting is the use of CALIOP for complementing the profiling done by the flights and check coherence among datasets, but not comparison. Therefore, please reorganize section 5 in this sense.*

We understand the reviewer's concern. First of all, we should recall that the CALIOP overpasses were chosen based on the analysis of the transport of the air masses. We therefore believe that the same type of aerosol is considered for the CALIOP overpass analysis and the analysis of airborne lidar data. Nevertheless, changes in aerosol properties are always possible during transport and we agree that the validation of CALIOP data is still questionable, whereas several validation campaigns using direct measurement of extinction have been published, although not in Siberia. In the newly prepared version, the discussion of CALIOP overpasses will be used only to obtain additional information on aerosol distribution and optical properties at the regional scale during the aircraft campaigns.

---

## Author Comment (AC3) · 15 Jul 2020

We decided to withdraw the current manuscript in order to refocus the results of the airborne campaign towards the analysis of the Siberian aerosol sources observed by the aircraft and to drop the discussion on the lidar ratio characterization and comparison with CALIOP. Satellite data will be only provided as a mean to document the regional context of the airborne observations.

Some quick answers to the reviewer comments are nevertheless provided :

*1. The extinction-to-backscatter ratio (lidar ratio) of different aerosol types is one of the main products of this study. However, it is not measured directly, but retrieved using the backscatter coefficient from the airborne lidar and the aerosol optical depth (AOD) from the satellite (MODIS). The scientific advance by this indirect retrieval of the lidar ratio is little. And as the lidar ratio is not just a side product but the main product of the aerosol characterization in this study, it is not sufficiently well constrained.*

Yes we agree it is a difficult task to address this question without a direct measurement of the extinction profile. We have estimated the uncertainty in the retrieval when using a constraint with MODIS AOD by using the distribution of several MODIS AOD in the air masses sampled by the aircraft. However it is true that the spatio-temporal differences between the satellite observations and the aircraft observations is always a critical question, especially when the aerosol optical properties rapidly change with mixing or relative humidity evolution. The time difference between the MODIS observations and the aircraft observations are less than 1 hour for three cases (dusty mix, fresh fire and Siberian city emissions), while it is indeed between 4-5 h for the other three cases (aged fire, gas flaring, aged urban emissions from China).  In the new version of the paper the retrieval of an aircraft extinction profile based on a constraint by MODIS will be left out. The retrieval of an AOD below the aircraft will be made using our analysis of the type of aerosol based on  the FLEXPART analysis coupled with the in-situ measurements, while the lidar ratio for the corresponding aerosol type will be taken according to the existing values of the scientific literature. The MODIS AOD from Aqua and Terra will be only an additionnal information in the analysis of the airborne lidar data to provide the regional context of the aerosol distribution.

*2. The comparison of the airborne observations with the spaceborne CALIOP measurements is highly uncertain. The temporal and spatial distance is too large to draw valid conclusions.  The conclusions drawn with respect to CALIOP might be correct but are not based on a convincing comparison.*

We understand the reviewer's concern. First of all, we should recall that the CALIOP overpasses were chosen based on the analysis of the transport of the air masses. We therefore believe that the same type of aerosol is considered for the CALIOP overpass analysis and the analysis of airborne lidar data. Nevertheless, changes in aerosol properties are always possible during transport and we agree that the validation of CALIOP data is still questionable, whereas several validation campaigns using direct measurement of extinction have been published, altough not in Siberia. In the newly prepared version, the discussion of CALIOP overpasses will be used only to obtain additional information on aerosol distribution and optical properties at the regional scale during the aircraft campaigns.

*3. The depolarization ratio conveys important information about the particle shape and is therefore a key parameter in the aerosol classification presented in Burton et al., 2013, to which the authors refer in the manuscript. I wonder how the authors assign one of Burton's aerosol types in Tab. 1.*

We agree that the depolarization conveys important information. In this campaign it is not available. I think the reviewer misunderstood that the aerosol type classification proposed in this work is only based on the joint analysis of the FLEXPART simulations, satellite data and in-situ aircraft measurement. The airborne lidar is not used to derive the aerosol type.

*4. Two out of the six cases presented have significant ambiguities in the current version of the manuscript. Case 2 (Ob Valley gas flaring emissions) is a multilayer scenario. Case 6 (Long-range transport of Northern China emissions) the source appointment is not very convincing. The FLEXPART backward simulation (Fig. 18a) shows a large residence time in the area south of Yakutsk.*

For case two yes the satellite data and FLEXPART analysis suggests that dust aerosol layer might be encountered just above the aircraft. Although it is well separated from the gas flaring signature it is

true that a small additionnal contribution of this layer will be included in the MODIS AOD and not in the airborne lidar AOD. According to the CALIOP data analysis it will not be larger than 0.05. For case six, we agree the figure 18 is misleading. It is the average of the FLEXPART backward simulation between 0 and 5 km and the contribution of the Harbin region is smoothed out. If the FLEXPART simulation is limited to the aerosol layer detected between 2.5-4 km, the contribution of Harbin becomes quite obvious (see FLEXPART Potential Emission Sensitivity plot when splitting the release area for the 1km-2.5 km and 2.5km-4km altitude ranges). This will be discussed in the corrected version.

[Figure]

Fig. : FLEXPART potential emission sentivity map in second using 4 days backward simulations between July 16th 2013 and July 19th 2013. The black cross is the aerosol layer position observed by the airborne lidar in the 1-2 km (left) and 2.5-4km (right) altitude range. The Harbin location is at 45°N, 126°E.

*I would encourage the authors to focus on different aspects of their campaign: The differences between fresh and aged forest fire smoke or between gas flaring emissions and urban pollution. Also, a stronger focus could be put on the comparison between the lidar and the in situ measurements.*

We thank the reviewer for this constructive comment. We decided to refocus the paper along these lines.